# Hypoxia triggers collective aerotactic migration in *Dictyostelium discoideum*

Olivier Cochet-Escartin[1†*], Mete Demircigil[2†], Satomi Hirose[3,4], Blandine Allais[1], Philippe Gonzalo[5,6], Ivan Mikaelian[5], Kenichi Funamoto[3,4,7], Christophe Anjard[1], Vincent Calvez[2†*], Jean-Paul Rieu[1†*]

[1]Institut Lumière Matière, UMR5306, Université Lyon 1-CNRS, Université de Lyon, Villeurbanne, France; [2]Institut Camille Jordan, UMR5208, Université Lyon 1-CNRS, Université de Lyon, Villeurbanne, France; [3]Graduate School of Biomedical Engineering, Tohoku University, Sendai, Japan; [4]Institute of Fluid Science, Tohoku University, Sendai, Japan; [5]Centre Léon Bérard, Centre de recherche en cancérologie de Lyon, INSERM 1052, CNRS 5286, Université Lyon 1, Université de Lyon, Lyon, France; [6]Laboratoire de Biochimie et Pharmacologie, Faculté de médecine de Saint-Etienne, CHU de Saint-Etienne, Saint-Etienne, France; [7]Graduate School of Engineering, Tohoku University, Sendai, Japan

**\*For correspondence:**
olivier.cochet-escartin@univ-lyon1.fr (OC-E);
Vincent.Calvez@math.cnrs.fr (VC);
jean-paul.rieu@univ-lyon1.fr (J-PR)

[†]These authors contributed equally to this work

**Competing interests:** The authors declare that no competing interests exist.

**Abstract** Using a self-generated hypoxic assay, we show that the amoeba *Dictyostelium discoideum* displays a remarkable collective aerotactic behavior. When a cell colony is covered, cells quickly consume the available oxygen ($O_2$) and form a dense ring moving outwards at constant speed and density. To decipher this collective process, we combined two technological developments: porphyrin-based $O_2$-sensing films and microfluidic $O_2$ gradient generators. We showed that *Dictyostelium* cells exhibit aerotactic and aerokinetic response in a low range of $O_2$ concentration indicative of a very efficient detection mechanism. Cell behaviors under self-generated or imposed $O_2$ gradients were modeled using an in silico cellular Potts model built on experimental observations. This computational model was complemented with a parsimonious 'Go or Grow' partial differential equation (PDE) model. In both models, we found that the collective migration of a dense ring can be explained by the interplay between cell division and the modulation of aerotaxis.

## Introduction

Oxygen is the main electron acceptor for aerobic organism to allow efficient ATP synthesis. This high-energy metabolic pathway has contributed to the emergence and diversification of multicellular organism (*Chen et al., 2015*). While high $O_2$ availability in the environment seems a given, its rapid local consumption can generate spatial and temporal gradients in many places, including within multicellular organism. Oxygen level and gradients are increasingly recognized as a central parameter in various physiopathological processes (*Tonon et al., 2019*), cancer and development. The well-known HIF (hypoxia-inducible factor) pathway allows cells to regulate their behavior when exposed to hypoxia. At low $O_2$ levels, cells accumulate HIFα leading to the expression of genes that support cell functions appropriate to hypoxia (*Pugh and Ratcliffe, 2017*).

Another strategy used by organisms facing severe oxygen conditions is to move away from hypoxic regions, a mechanism called aerotaxis and first described in bacteria (*Engelmann, 1881*; *Winn et al., 2013*). Aerotaxis will occur at the interface between environments with different oxygen content, such as soil/air, water/air or even within eukaryotic multicellular organisms between different tissues (*Lyons et al., 2014*). In such organisms, oxygen was proposed to be a morphogen as in placentation (*Genbacev et al., 1997*) or a chemoattractant during sarcoma cell invasion

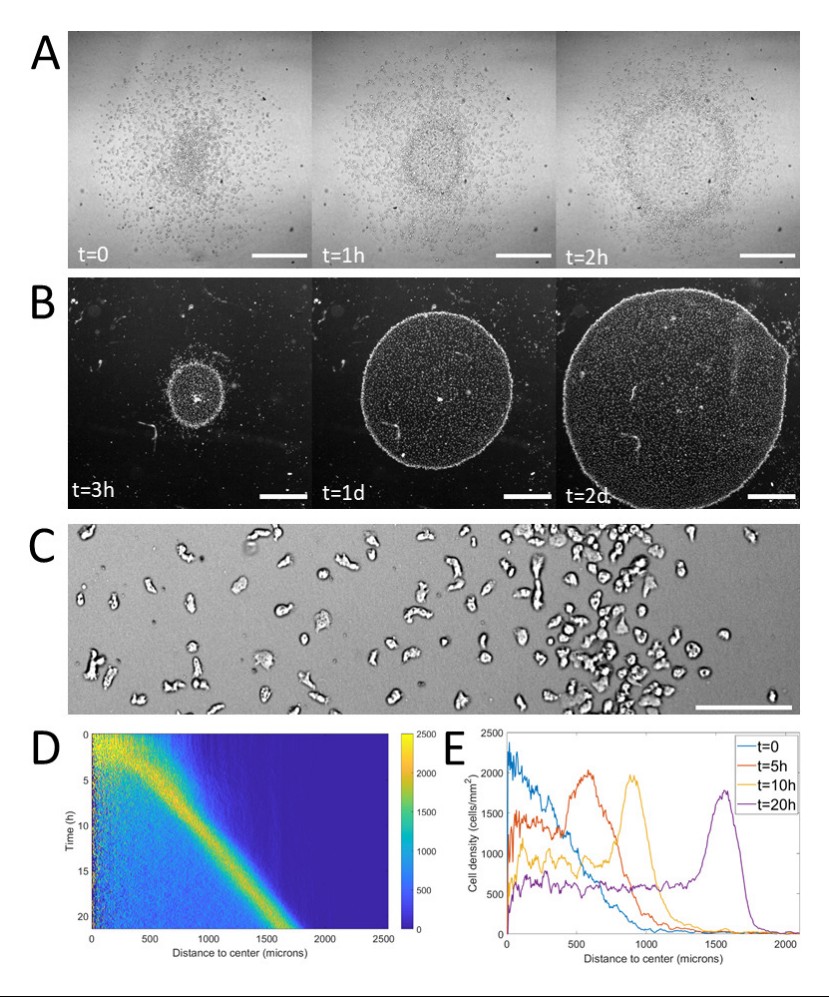

**Figure 1.** Formation and dynamics of a dense ring of cells after vertical confinement. (**A**) Snapshots of early formation, scale bars: 500 µm. (**B**) Snapshots at longer times imaged under a binocular, scale bars: 1 mm. (**C**) Close up on a ring (band with a higher density on the right hand side) already formed moving rightward and showing different cellular shapes in the ring and behind it, scale bar: 100 µm. (**D**) Kymograph of cell density over 20 hr showing the formation and migration of the highly dense ring. (**E**) Cell density profiles in the radial direction at selected time points.

The online version of this article includes the following source data and figure supplement(s) for figure 1:

**Source data 1.** Raw data for *Figure 1*.
**Figure supplement 1.** Measurement of the confinement height 105 min after covering a cell colony with ~1000 cells plated on plastic with a coverglass using.
**Figure supplement 2.** Ring formation time decreases with cell number.
**Figure supplement 2—source data 1.** Raw data for *Figure 1—figure supplement 2*.
**Figure supplement 3.** Morphological properties of a propagating ring.
**Figure supplement 3—source data 1.** Raw data for *Figure 1—figure supplement 3*.
**Figure supplement 4.** Effective cell diffusion constant and instantaneous speeds as a function of distance to the center.
**Figure supplement 4—source data 1.** Raw data for *Figure 1—figure supplement 4*.
**Figure supplement 5.** Cell velocity bias in the spot assay as a function of distance to the center.
**Figure supplement 5—source data 1.** Raw data for *Figure 1—figure supplement 5*.

(*Lewis et al., 2016*). Aerotaxis may also play a role in morphogenesis. The notion that gradients of $O_2$ and energy metabolism govern spatial patterning in various embryos dates back to the classic work of *Child, 1941*. Such notions have mostly been abandoned due to the inability to visualize such a gradient or clarify whether they are the result or the cause of developmental patterning

(*Coffman and Denegre, 2007*). Even at the single-cell level, *in vitro* experimental studies on aerotaxis are rare. One reason might be technical: gradient control and live monitoring of oxygen concentrations at the cellular level are difficult. More recently, Chang et al. found an asymmetric distribution of hypoxia-inducible factor regulating dorsoventral axis establishment in the early sea urchin embryo (*Chang et al., 2017*). Interestingly, they also found evidence for an intrinsic hypoxia gradient in embryos, which may be a forerunner to dorsoventral patterning.

Self-generated chemoattractant gradients have been reported to trigger the dispersion of melanoma cells (*Muinonen-Martin et al., 2014*; *Stuelten, 2017*), *Dictyostelium* cells (*Tweedy et al., 2016*) or the migration of the zebrafish lateral line primordium (*Donà et al., 2013*; *Venkiteswaran et al., 2013*). The mechanism is simple and very robust: the cell colony acts as a sink for the chemoattractant, removes it by degradation or uptake creating a gradient that, in turn, attracts the cells as long as the chemoattractant is present in the environment. Physiologically speaking, self-generated gradients have been demonstrated to increase the range of expansion of cell colonies (*Cremer et al., 2019*; *Tweedy and Insall, 2020*) and to serve as directional cues to help various cell types navigate complex environments, including mazes (*Tweedy et al., 2020*). Recently, it was demonstrated that after covering an epithelial cell colony by a coverglass non permeable to $O_2$, peripheral cells exhibit a strong outward directional migration to escape hypoxia from the center of the colony (*Deygas et al., 2018*). This is a striking example of a collective response to a self-generated oxygen gradient by eukaryotic cells. Oxygen self-generated gradients could therefore play important roles in a variety of contexts, such as development, cancer progression, or even environmental navigation in the soil.

*Dictyostelium discoideum* (*Dd*) is an excellent model system to study the fairly virgin field of aerotaxis and of self-generated gradients. *Dd* is an obligatory aerobic organism that requires at least 5% $O_2$ to grow at optimal exponential rate (*Cotter and Raper, 1968*; *Sandonà et al., 1995*) while slower growth can occur at 2% $O_2$. However, its ecological niche in the soil and around large amount of bacteria can result in reduced $O_2$ availability. During its multicellular motile stage, high oxygen level is one of the signal used to trigger culmination of the migrating slug (*Xu et al., 2012*). For many years, *Dd* has been a classical organism to study chemotaxis and has emulated the development of many models of emergent and collective behavior since the seminal work of Keller and Segel (*Hillen and Painter, 2009*; *Keller and Segel, 1970*). An integrated approach combining biological methods (mutants), technological progress, and mathematical modeling is very valuable to tackle the issue of aerotaxis.

In this article, we study the influence of $O_2$ self-generated gradients on *Dd* cells. Using a simple confinement assay, microfluidic tools, original oxygen sensors and theoretical approaches, we show that oxygen self-generated gradients can direct a seemingly collective migration of a cell colony. Our results confirm the remarkable robustness and long-lasting effect of self-generated gradients in collective migration. This case where oxygen is the key driver also suggests that self-generated gradients are widespread and a possible important feature in multicellular morphogenesis.

## Results

### Confinement triggers formation and propagation of a self-sustained cell ring

To trigger hypoxia on a colony of *Dd* cells, we used a vertical confinement strategy (*Deygas et al., 2018*). A spot of cells with a radius of about 1 mm was deposited and covered by a larger glass coverslip with a radius of 9 mm. We measured the vertical confinement through confocal microscopy and found the

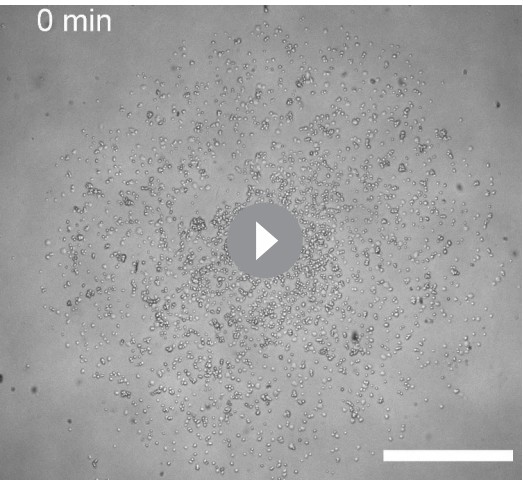

**Video 1.** Initial phase (0–4 hr) of ring formation and migration. Scale bar: 500 µm.
https://elifesciences.org/articles/64731#video1

height between the bottom of the plate and the coverslip to be 50 μm (*Figure 1—figure supplement 1*).

Using spots containing around 2000 cells (initial density around $10^3$ cells/mm$^2$), the formation of a dense ring of cells moving outwards was observed as quickly as 30 min after initiation of the confinement (*Figure 1A* and *Video 1*). This formation time however depended non-linearly on initial cell density (the denser, the faster, *Figure 1—figure supplement 2*). Once triggered, this collective migration was self-maintained for tens of hours, even days and the ring could, at these points, span centimeters (*Figure 1B*).

Notably, as the ring expanded outwards, it left a trail of cells behind. This led to the formation of a central zone populated by cells which did not contribute directly to the migration of the ring (*Figure 1B*) but were still alive and moving albeit a clear elongated phenotype resembling pre-aggregative *Dd* cells (*Figure 1C* and *Video 2*). In comparison, cells in the ring or outside the colony were rounder, as usual vegetative cells (*Delanoë-Ayari et al., 2008*).

To study the properties of the ring, we computed density profiles using radial coordinates from the center of the colony to study cell density as a function of time and distance to the center (*Figure 1D–E*). We found that after a transitory period corresponding to the ring passing through the initial spot, the density in the ring, its width and its speed all remained constant over long time scales (*Figure 1—figure supplement 3*). The speed and density of the ring were found to be 1.2 ± 0.3 μm/min (mean ± std, N=9 independent experiments) and 1.9 10$^3$ ± 0.3 10$^3$ cells/mm$^2$ (mean ± std, N=4 independent experiments, that is fourfold higher than behind it, *Figure 1E*) respectively. The density of cells left behind the ring was also found to remain constant after a transient regime (*Figure 1D*). As the diameter of the ring increased over time, the absence of changes in morphology implies an increase of the number of cells and thus an important role of cell division.

Overall, this self-sustained ring propagation is very robust and a long lasting collective phenotype that can easily be triggered experimentally. This shows that the spot assay is an excellent experimental system to study the response of a variety of cell types to vertical confinement and its physiological consequences (*Deygas et al., 2018*).

## Cell dynamics in different regions

Following the reported shape differences, we questioned how cells behaved dynamically in different regions. To do so, we performed higher resolution, higher frame rate experiments to allow cell tracking over times on the order of tens of minutes. Both the cell diffusion constant and instantaneous cell speeds were fairly constant throughout the entire colony (*Figure 1—figure supplement 4*). Cell diffusion was 28.2 ± 1.4 μm$^2$/min (N=3 independent experiments, each containing at least 2000 cells), comparable to our measurement of activity at very low oxygen level in the microfluidic device (see below). To test the influence of motion bias, we projected cell displacements on the radial direction and computed mean speeds in this direction as a function of distance to the center. Random motion, either persistent or not, would lead to a null mean radial displacement whereas biased migration would be revealed by positive (outward motion) or negative (inward motion) values. Here, we found that significantly non-zero biases were observed only in a region spanning the entire ring and a few tens of microns behind and in front of it with the strongest positive biases found in the ring (*Figure 1—figure supplement 5*).

Overall, our results show that the different regions defined by the ring and its dynamics can be characterized in terms of cell behavior: (i) behind the ring in the hypoxic region: elongated

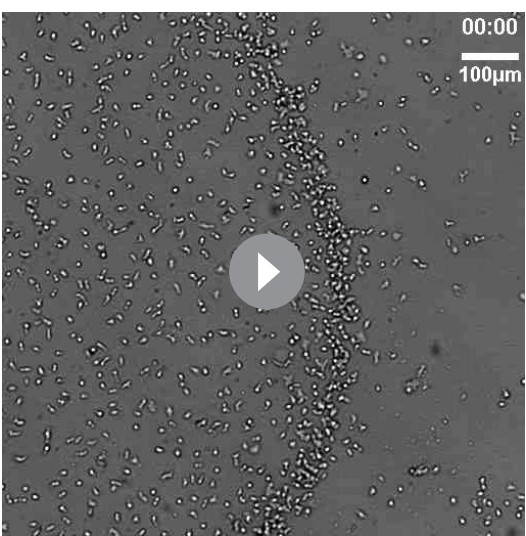

**Video 2.** High framerate, high-resolution imaging of cell dynamics in and behind the ring over 15 min. Time is in min:s and the scale bar represents 100 μm.
https://elifesciences.org/articles/64731#video2

shape, normal speeds, and low bias; (ii) in the ring: round shape, normal speeds and high bias.

## Response of *Dd* cells to controlled oxygen gradients

The spot assay is experimentally simple but is not ideally suited to decipher the response of *Dd* cells to hypoxia since local concentrations and gradients of oxygen are coupled to cell dynamics and thus very difficult to manipulate. We hence designed a new double-layer PDMS microfluidic device allowing to quickly generate oxygen gradients in a continuous, controlled manner (*Figure 2A*). Briefly, cells were seeded homogenously within a media channel positioned 500 µm below two gas channels continuously perfused with pure nitrogen on one side and air on the other. As PDMS is permeable to these gases, the gas flows imposed hypoxic conditions on one side of the media channel while the other was kept at atmospheric oxygen concentration. Of note, the distance between the two gas channels, thereafter called the gap, varied from 0.5 mm to 2 mm in order to modify the steepness of the gradients in the median region of the media channels (*Figure 2A* and Materials and methods).

To make sure that the gas flows were sufficient to maintain a constant $O_2$ distribution against leakages and against small variation in the fabrication process, we also developed $O_2$-sensing films to be able to experimentally measure $O_2$ profiles both in the microfluidic devices and in the spot assay. These films consisted of porphyrin based $O_2$ sensors embedded in a layer of PDMS. As $O_2$ gets depleted, the luminescence quenching of the porphyrin complex is reduced leading to an increase in fluorescence intensity (*Ungerböck et al., 2013*). Quantitative oxygen measurements were then extracted from this fluorescent signal using a Stern-Volmer equation (see Materials and methods and *Figure 2—figure supplements 1–4* for details).

Within 15 min, we observed the formation of a stable $O_2$ gradient in the devices closely resembling numerical predictions with or without cells (*Figure 2B* and *Figure 2—figure supplements 5–7*).

We then turned our attention to the reaction of the cells to this external gradient. We first noticed that depending on local $O_2$ concentrations, cell motility was remarkably different. Using cell tracking, we found that cell trajectories seemed much longer and more biased in hypoxic regions (*Figure 2C*). These aerokinetic (large increase in cell activity) and aerotactic responses were confirmed by quantifying the mean absolute distance travelled by cells (*Figure 2D* top), or the mean distance projected along the gradient direction (*Figure 2D* bottom) in a given time as a function of position in the device (*Figure 2D*). Since cells in the microfluidic devices were also experiencing oxygen gradients, we further tested if the observed was true aerokinesis. To do so, we compared cell motility in homogenous environments of either 20.95% or 0.4% $O_2$. We found cell diffusion constant to be D=40.2±9.6 µm²/min (mean ± std) at 0.4% (*Figure 2—figure supplement 8*), comparable to our measurements in the center of the spot (*Figure 1—figure supplement 4*). At atmospheric oxygen concentrations though, this effective diffusion was clearly reduced as we measured it to be D=19.2±8.8 µm²/min (*Figure 2—figure supplement 8*). The very significant difference (p<0.0001) demonstrates that *Dd* cells show an aerokinetic positive response to low oxygen, even in the absence of gradients. The second important observation stemming from the microfluidic experiments is an accumulation of cells at some midpoint within the cell channel (*Figure 2E*). Naively, one could have expected cells to follow the $O_2$ gradient over its entire span leading to an accumulation of all cells on the $O_2$ rich side of the channel. This did not happen and, instead, cells seemed to stop responding to the gradient at a certain point. Similarly, we observed a strong positive bias in hypoxic regions but the bias quickly fell to 0 as cells moved to oxygen levels higher than about 2% (*Figure 2D*), confirming that the observed cell accumulation was a result of differential migration and not, for example, differential cell division. In addition, if we inverted the gas channels halfway through the experiment, we observed that the cells responded in around 15 min (which is also the time needed to re-establish the gradient, see *Figure 2—figure supplement 6*) and showed the same behavior, albeit in reverse positions. We measured the bias for the different gaps and for the situation of reversed gradient and obtained a value of 1.1 ± 0.4 µm/min (N=6, three independent experiments and for each, both directions of the gradient, each value stemming from a few hundred cells).

Of note, the position at which cells accumulated and stopped responding to the gradient was still in the region were the gradient was constantly increasing. This led to the hypothesis that, in addition

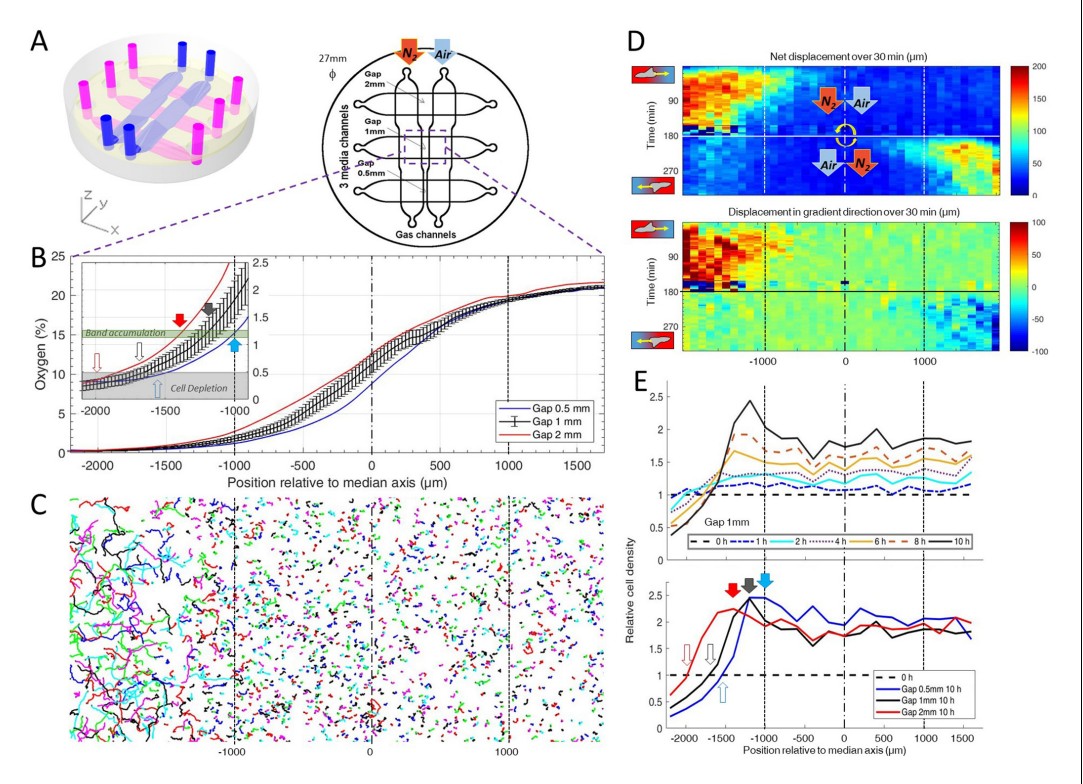

**Figure 2.** *Dictyostelium* single cells are attracted by an external $O_2$ gradient when $O_2$ level drops below 2%. (**A**) Schemes of the new double-layer PDMS microfluidic device allowing the control of the $O_2$ gradient by the separation distance (gap) between two gas channels located 0.5 mm above the three media channels and filled with pure nitrogen, and air (21% $O_2$). (**B**) Measured $O_2$ concentration profiles 30 min after $N_2$-Air injection to the left and right channels respectively (0–21% gradient) as a function of the position along the media channel for the three gaps. Error bars (see Methods) are reported only for gap 1 mm for clarity. The inset shows the 0–2.5% region under the nitrogen gas channel (arrows, see E). (**C**) Trajectories lasting 1 hr between 3 hr and 4 hr after establishment of a 0–21% gradient. Cells are fast and directed toward the air side in the region beyond the −1000 µm limit ($O_2$<2%). (**D**) Cell net displacement over 30 min (end to end distance, top kymograph) and 30 min displacement projected along gradient direction (bottom kymograph). Cells are fast and directed toward $O_2$, where $O_2$<2%, within 15 min after 0–21% gradient establishment at t=0. At t=180 min, the gradient is reversed to 21–0% by permuting gas entries. Cells within 15 min again respond in the 0–2% region. (**E**) Relative cell density histogram (normalized to t=0 cell density) as a function of the position along media channel. Top panel: long term cell depletion for positions beyond −1600 µm ($O_2$<0.5%, see inset of B) and resulting accumulation at about −1200 µm for gap 1 mm channel. The overall relative cell density increase is due to cell divisions. Bottom panel: cell depletion and accumulation at 10 hr for the three gaps. The empty and filled arrows pointing the limit of the depletion region, and max cell accumulation respectively are reported in the inset of B.

The online version of this article includes the following source data and figure supplement(s) for figure 2:

**Source data 1.** Raw data for *Figure 2*.
**Figure supplement 1.** Oxygen profile measurements inside the microfluidic gradient generator device with a sensing film mounted on the bottom of the media channel.
**Figure supplement 1—source data 1.** Raw data for *Figure 2—figure supplement 1*.
**Figure supplement 2.** Typical calibration data of sensing films mounted on a microfluidic device.
**Figure supplement 2—source data 1.** Raw data for *Figure 2—figure supplement 2*.
**Figure supplement 3.** Typical calibration data and oxygen profile measurement with covered sensing films for the spot assay.
**Figure supplement 3—source data 1.** Raw data for *Figure 2—figure supplement 3*.
**Figure supplement 4.** Image analysis pipeline to quantify oxygen map from $O_2$ sensitive sensing films.
**Figure supplement 5.** Numerical simulations of oxygen profiles.
**Figure supplement 5—source data 1.** Raw data for *Figure 2—figure supplement 5*.
**Figure supplement 6.** Experimental oxygen gradient establishment in the microfluidic device (gap 0.5 mm).
**Figure supplement 6—source data 1.** Raw data for *Figure 2—figure supplement 6*.
**Figure supplement 7.** Influence of plated cells on the steady oxygen tension in the microfluidic device (Computational results).
**Figure supplement 7—source data 1.** Raw data for *Figure 2—figure supplement 7*.
**Figure supplement 8.** Aerokinesis of *Dd* cells in homogenous environments.
**Figure supplement 8—source data 1.** Raw data for *Figure 2—figure supplement 8*.

to gradient strength, $O_2$ levels also play an important role in setting the strength of aerotaxis displayed by *Dd* cells.

Furthermore, when we compared experiments performed with different gaps, we found that the position of cell accumulation varied from one channel to another (*Figure 2E*). However, our $O_2$ sensors indicated that the accumulation occurred at a similar $O_2$ concentration of about 1% in all three channels (inset of *Figure 2B*) thus strongly hinting that the parameter controlling the aerotactic response was $O_2$ levels.

Overall, these experiments in controlled environments demonstrated two main features of the response of *Dd* cells to hypoxia: a strong aerokinetic response and a positive aerotactic response, both modulated by local $O_2$ levels regardless of the local gradient. These results reveal a subtle cross talk between $O_2$ concentrations and gradients in defining cell properties and it would be very informative, in the future, to study in details the reaction of *Dd* cells to various, well defined hypoxic environments where $O_2$ concentrations and gradients can be independently varied.

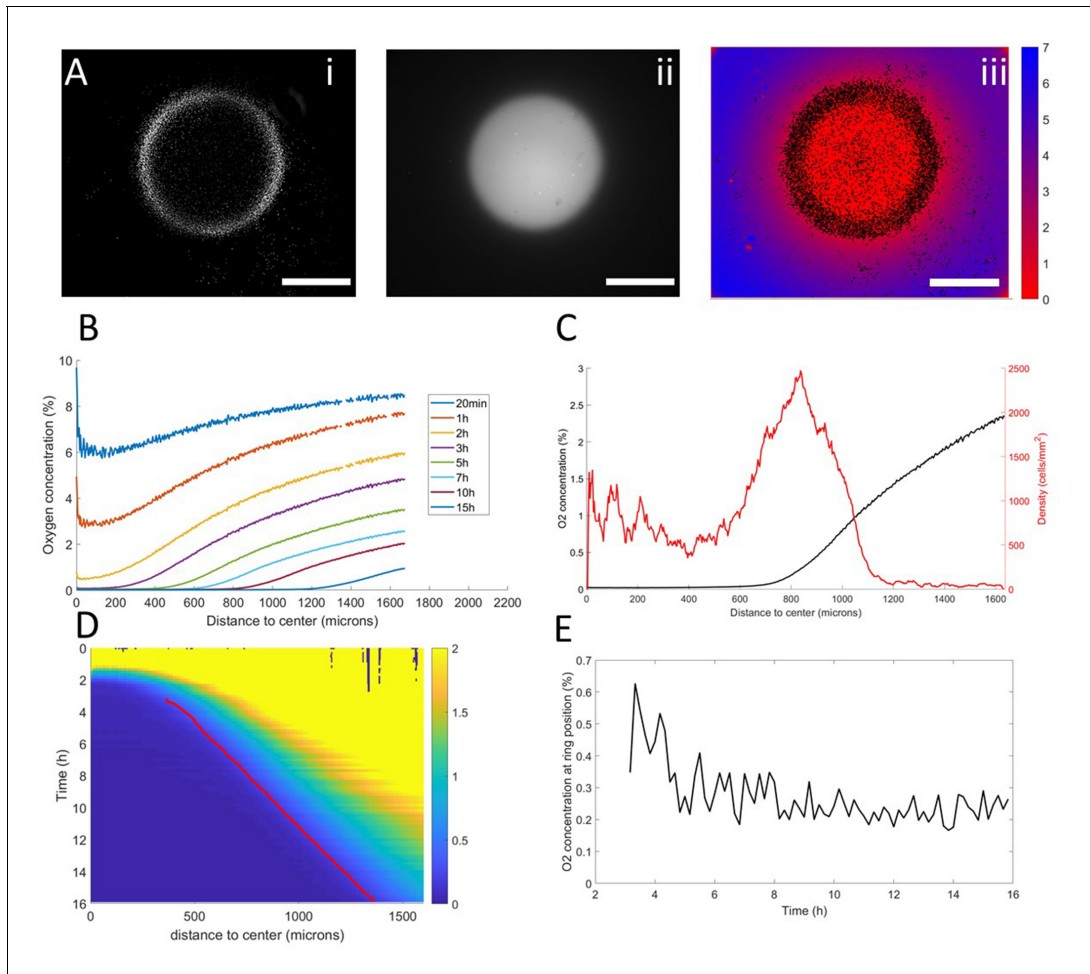

**Figure 3.** Interplay between ring dynamics and $O_2$ profiles. (A) (i) Treated image showing cell distribution at t=10h, (ii) raw fluorescent signal indicative of strong $O_2$ depletion, (iii) reconstructed image showing the center of mass of all detected cells and quantitative $O_2$ profiles (colorbar, in % of $O_2$), scale bars: 1 mm. (B) $O_2$ profiles averaged over all angles and shown at different times. (C) Radial profile of cell density and $O_2$ concentration at t=10h showing the position of the ring relative to the $O_2$ profile. (D) Kymograph of $O_2$ concentration (colorbar in %) with the position of the ring represented as a red line. The colormap is limited to the 0–2% range for readability but earlier time points show concentrations higher than the 2% limit. (E) $O_2$ concentration as measured at the position of the ring as a function of time showing that the ring is indeed following a constant concentration after a transitory period.

The online version of this article includes the following source data for figure 3:

**Source data 1.** Raw data for *Figure 3*.

## Coupled dynamics between oxygen profiles and collective motion

Thanks to these results, we turned our attention back to the collective migration of a ring of cells and asked whether similar aerotactic behaviors were observed under self-generated gradients. To do so, we performed spot experiments on the $O_2$-sensing films described above which allowed us to image, in parallel, cell behavior and $O_2$ distribution (*Figure 3A*, *Figure 2—figure supplement 3* and *Video 3*).

In a first phase, preceding the formation of the ring, cell motion was limited and the structure of the colony remained mostly unchanged. As $O_2$ was consumed by cells, depletion started in the center and sharp gradients appeared at the edges of the colony (*Figure 3B–C*).

Then, the ring formed and started moving outwards, $O_2$ depletion continued and the region of high $O_2$ gradients naturally started moving outwards (*Figure 3B*). At this point, coupled dynamics between the cells and the $O_2$ distribution appeared and we observed that the position of the ring closely followed the dynamics of the $O_2$ field (*Figure 3D*), that is it followed a constant concentration of oxygen of 0.25% (*Figure 3C*).

In the process, three distinct regions were created. Behind the ring, $O_2$ was completely depleted and thus no gradient was visible. In front of the ring, the $O_2$ concentration remained high with high gradients. Finally, in the ring region, $O_2$ was low (<1%) and the gradients were strong. Based on our results in externally imposed gradients, we would thus expect cells to present a positive aerotactic bias mostly in the ring region which is indeed what we observed (*Figure 1—figure supplement 5*).

## Minimal cellular Potts model

Based on these experimental results, we then asked whether this subtle response of *Dd* cells to complex oxygen environments was sufficient to explain the emergence of a highly stable, self-maintained collective phenomenon. To do so, we developed cellular Potts models based on experimental observations and tested whether they could reproduce the observed cell dynamics. Briefly, the ingredient underlying the model are as follows (details can be found in the Materials and methods section). First, all cells consume the oxygen that is locally available at a known rate (*Torija et al., 2006*). Cell activity increases at low $O_2$. Cells respond positively to $O_2$ gradients with a modulation of the strength of this aerotaxis based on local $O_2$ concentrations, as observed in our microfluidic experiments. Finally, all cells can divide as long as they sit in a high enough $O_2$ concentration (chosen at 0.7%) since it was demonstrated that cell division slows down in hypoxic conditions (*Schiavo and Bisson, 1989*; *West et al., 2007*). Of note, all parameters were scaled so that both time and length scales in the Potts models are linked to experimental times and lengths (see Materials and methods).

Although this model is based on experimental evidence, some of its parameters are not directly related to easily measurable biological properties. Therefore, we decided to fit our parameters to reproduce as faithfully as possible the results of our microfluidic experiments. Through a trial and error procedure, we managed to reproduce these results qualitatively and quantitatively (*Video 4*) in terms of collective behavior, cell accumulation, and individual cell behavior (*Figure 4—figure supplement 1*).

We then applied this model and added $O_2$ consumption by cells, with initial conditions mimicking our spot assay and other ingredients mimicking the vertical confinement. We observed the rapid formation and migration of a ring (*Figure 4A–B*, *Video 5*). This ring was remarkably similar to that observed in experiments. In particular, we found its speed to be constant after an initial transitory period (*Figure 4C*,

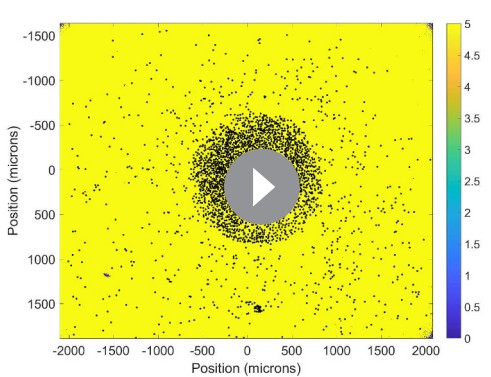

**Video 3.** Reconstruction of cell and oxygen dynamics from a spot experiment on an oxygen sensor. Cell positions are shown as black dots, oxygen in colors (scale bar in %). The entire movie spans 15 hr of experiment.

https://elifesciences.org/articles/64731#video3

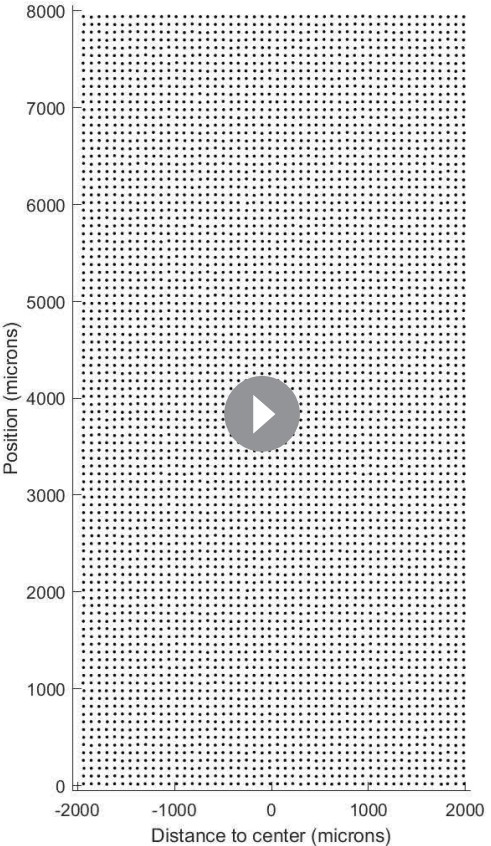

**Video 4.** Dynamics of the Potts model reproducing microfluidic experiments. Low oxygen regions are on the left and high oxygen on the right. Cell positions are shown as black dots and the entire movie represents the equivalent of 10 hr of experiments.
https://elifesciences.org/articles/64731#video4

*Figure 4—figure supplement 2*). This speed was also comparable to experimental ones. Similarly, the morphology of these simulated rings was constant over time with a fixed cell density and width (*Figure 4—figure supplement 2*). Finally, cell behavior was qualitatively well reproduced by this model (*Figure 4—figure supplement 3*).

In terms of coupled dynamics between cell density and $O_2$ profiles, we found here too that the driving force behind this collective phenomenon was the fact that the ring followed a constant $O_2$ concentration (*Figure 4D–E*).

We then asked what were the key ingredients in the model to trigger this phenomenon, a question we explored by tuning our original Potts model. We started by dividing cell consumption of oxygen by a factor of 3 (*Figure 5A*) and found that it did not significantly change the ring speed but could change the aspect of cell density in the central region. We then turned our attention to other key elements in the model.

If we turned off cell division in our models, the formation of the ring was mostly unchanged but after a short time, the ring started slowing down and even stopped as cell density was no longer sufficient to reach highly hypoxic conditions (comparing *Figure 5A and B*). Second, we asked whether the observed and modeled aerokinesis was necessary to reproduce the collective migration. We found that it wasn't as models ran at different effective temperatures applied to all cells regardless of local $O_2$ concentrations all showed qualitatively similar behavior (see for example *Figure 5C*). Of note though, lower effective temperatures led to less dense rings as fewer cells were able to start in the ring (*Figure 5—figure supplement 1*). Finally, we found that modulation of aerotaxis by local $O_2$ concentrations was essential. Indeed, as we increased the range of $O_2$ concentration at which aerotaxis is at play (*Figure 5G–H*), we found that forming rings became wider and less dense (*Figure 5D–E*) to the point where no actual ring could be distinguished if aerotaxis was kept constant for all cells (*Figure 5I*).

These numerical simulations based on cellular Potts models provide a good intuition of the phenomenon and reveal that cell division and aerotactic modulation are the two key ingredients to reproduce the ring of cells. Because of their versatility, they can also be used to make some predictions on the observed phenomenon. Experimentally, we tested two such predictions to demonstrate the relevance of the underlying assumptions.

First, we show in *Figure 5B* the effect of turning cell division off in the simulated spot. A similar result can be achieved by placing cells in a phosphate buffer medium, lacking nutrients and thus blocking cell division (*Kelly et al., 2021*). In this situation, at short time scales, a ring of cells started forming and expanding outwards in a similar fashion as in nutritive medium (*Figure 5—figure supplement 2*). After a few hours, however, the ring started slowing down until it completely stopped and cells started dispersing again. This is in complete agreement with the predictions of the Cellular Potts Model, as one can see by comparing the density kymographs (*Figure 5A* and *Figure 5—figure*

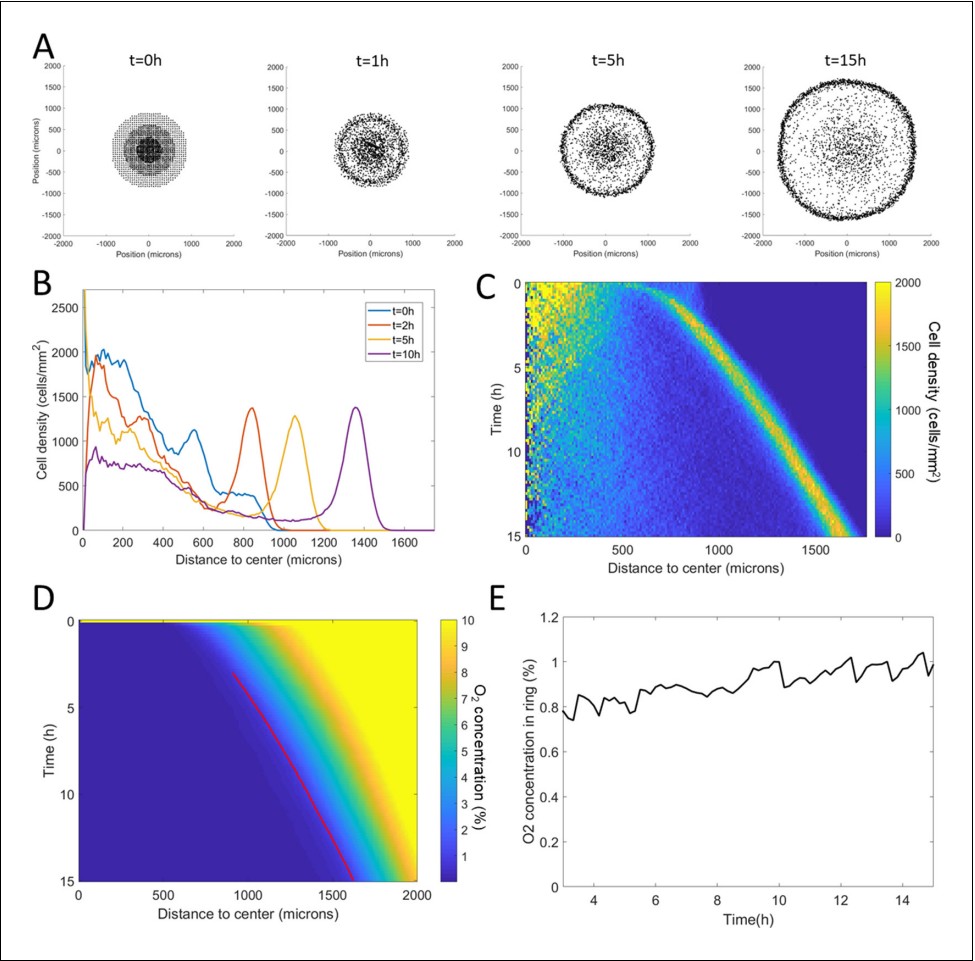

**Figure 4.** Minimal Potts model of ring formation and migration. (A) Snapshots of a simulated colony of cells showing the formation of highly dense ring of cells. (B) Cell density profiles averaged over all angles for four different times. (C) Corresponding kymograph of cell density (colorbar in cells/mm²) as a function of time and distance to the center. Quantification in terms of microns and hours is described in the Materials and methods section. (D) Kymograph of $O_2$ concentration (colorbar in %) with the position of the ring represented as a red line. The colormap is limited to the 0–10% range for readability but earlier time points show concentrations higher than the 10% limit. (E) $O_2$ concentration at the ring position as a function of time showing that, here too, the ring follows a constant $O_2$ concentration.

The online version of this article includes the following source data and figure supplement(s) for figure 4:

**Source data 1.** Raw data for *Figure 4*.

**Figure supplement 1.** Adjusting Potts model (right) to microfluidic experiments (left).

**Figure supplement 1—source data 1.** Raw data for *Figure 4—figure supplement 1*, experiments corresponding to the left column.

**Figure supplement 1—source data 2.** Raw data for *Figure 4—figure supplement 1*, simulations corresponding to the right column.

**Figure supplement 2.** Potts model ring features with parameters adjusted from the microfluidic experiments (*Figure 4—figure supplement 1*).

**Figure supplement 2—source data 1.** Raw data for *Figure 4—figure supplement 2*.

**Figure supplement 3.** Comparison of cell behavior in spot experiments (left) and Potts models (right).

**Figure supplement 3—source data 1.** Raw data for *Figure 4—figure supplement 3*, experiments corresponding to the left column.

**Figure supplement 3—source data 2.** Raw data for *Figure 4—figure supplement 3*, simulations corresponding to the right column.

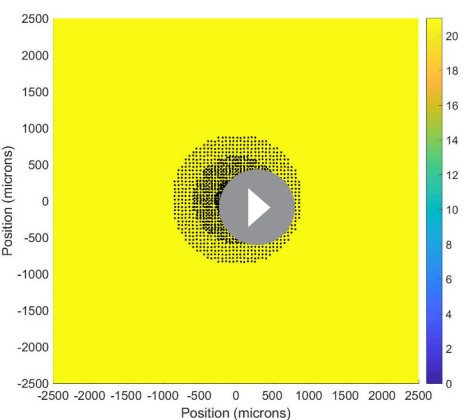

**Video 5.** Dynamics of the Potts model reproducing the spot experiments. Cell positions are shown as black dots and the oxygen is in colors (in %).
https://elifesciences.org/articles/64731#video5

supplement 2) and firmly demonstrates the importance of cell division in this behavior. At longer time scales though (t>10h), Dd cells started forming aggregates and entering a developmental phase (*Video 6*). This aggregation is presumably due to the concomitant expression of cell adhesion molecules and, apparition of self-organizing secreted cAMP pulses whose timing agrees with the one reported in classical free cell spot aggregation assays (*Gregor et al., 2010*). Cell-cell adhesion and cAMP signaling are not included in our models or numerical simulations that hence cannot predict the long times in *Video 6*. However, the timing is well separated from the end of the ring expansion period (t<3.5h). This still demonstrates that the phenomenon observed here is relevant for both the single cell and collective stages of Dd life cycle.

Second, we used these numerical simulations to predict the behavior of cells in more complex environments. One can see the expansion of the ring as a way for each cell to optimize its own resources. This begs the question of what happens when more than one colony is present in the environment, a problem more directly relevant for real life situations. Can the different colonies sense their respective presence and adapt accordingly by migrating preferably away from one another or, on the other hand, will the depletion of oxygen induced by a neighboring colony increase hypoxia on this side and therefore accelerate migration? In this case, what would happen when two rings come in contact? We started exploring this question by simulating two colonies put in close proximity. These simulations predict that the formed rings do not repel each other, instead they tend to rush toward one another and, when they meet, they fuse together to make an elliptical front which then relaxes towards a more circular shape (*Video 7*). We then performed the corresponding experiment and found very similar behavior (*Video 7*).

Overall, these results show that the cellular Potts model indeed recapitulates all the major experimental observations with only two key ingredients (cell division and aerotactic modulation). However, they fall short of giving an in-depth quantitative description because they rely on many parameters and are not amenable to theoretical analysis per se.

## 'Go or Grow' hypothesis: a Mean-field approach

In order to complement the methodology of the cellular Potts model, we developed a mean-field approximation of the latter: the cell density $\rho$ is subject to a reaction-advection-diffusion partial differential equation (PDE):

$$\frac{\partial \rho}{\partial t} = D\nabla \cdot (\nabla \rho) - \nabla \cdot (a(C, \nabla C)\rho) + r(C)\rho \qquad (1)$$

$C$ is the oxygen concentration, $a(C, \nabla C)$ corresponds to the aerotactic advection speed and $r(C)$ to the cell division rate. By assuming radial symmetry in agreement with the experiments, we propose $a(C, \nabla C) = a(C, \partial_r C) = \lambda_{aero}(C)\partial_r C$, where $\lambda_{aero}(C)$ is the already mentioned aerotactic strength fitting the microfluidic experiments with an upper $O_2$ concentration threshold $C_0$=0.7% (*Figure 5G* and Material and Methods) and $r(C) = \begin{cases} r_0, & \text{if } C > C_0 \\ 0, & \text{if } C < C_0 \end{cases}$ is the division rate. When not specified, we use the same threshold $C_0$ for cell division and aerotaxis as for the cellular Potts model. Below, this assumption is coined as the 'Go or Grow' hypothesis. We thereby revisited the 'Go or Grow' model for glioma cells (*Hatzikirou et al., 2012*) with the transition between division and directional motion being mediated by oxygen levels rather than cell density in the mentionned

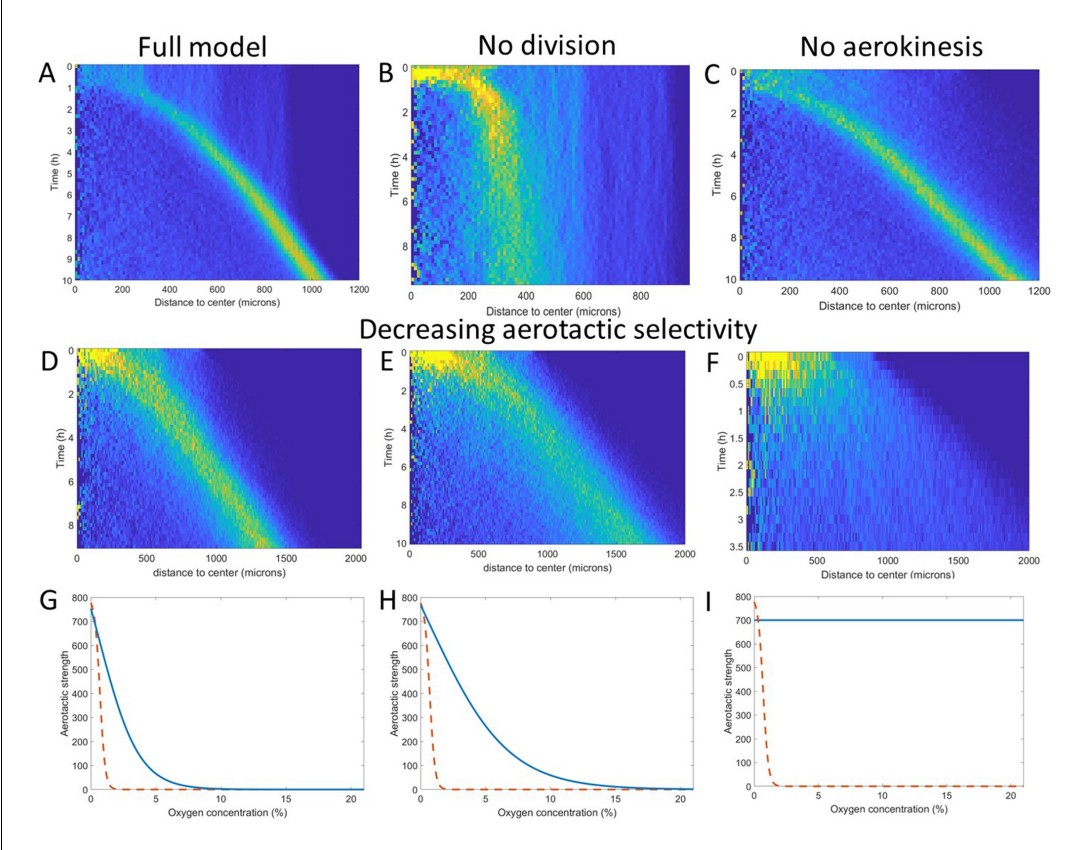

**Figure 5.** Key ingredients of the Potts model by density kymograph (DK) evaluation. (**A**) DK for the full model with reduced oxygen consumption as a basis for comparison. (**B**) DK in the absence of cell division, note the difference in length scale showing a clear limitation of motion in that case. (**C**) DK in the absence of aerokinesis (cell activity is not modulated by local oxygen concentrations). (**D**) DK with a modulation of aerotactic strength as shown in (**G**), note the wider ring. (**E**) DK with a modulation of aerotactic strength as shown in (**H**). (**F**) DK with a modulation of aerotactic strength as shown in (**I**), no ring appears and cells quickly migrate outwards as shown by the difference in time scales. (**G–I**) Three different aerotactic modulations, in blue, compared to the one used in the full model, shown in (**A**), drawn here as a red dashed line.

The online version of this article includes the following source data and figure supplement(s) for figure 5:

**Source data 1.** Raw data for *Figure 5*.
**Figure supplement 1.** Effect of temperature on ring migration in Potts models.
**Figure supplement 1—source data 1.** Raw data for *Figure 5—figure supplement 1*.
**Figure supplement 2.** Ring formation in a phosphate buffer.
**Figure supplement 2—source data 1.** Raw data for *Figure 5—figure supplement 2*.

study. Congestion effects such as they may arise in the cellular Potts model or in experiments have been ignored.

Oxygen is subject to a simple diffusion-consumption equation, with $b(C)$ the consumption rate of oxygen per cell (see Materials and methods):

$$\frac{\partial C}{\partial t} = D_{oxy} \nabla \cdot (\nabla C) - b(C)\rho \qquad (2)$$

The results obtained by numerical simulation of this mean-field model are comparable to the ones already obtained by the cellular Potts model: emergence of a high cell density area traveling at constant speed $\sigma \approx 1.0$ μm/min, leaving behind a trail of cells (*Figure 6A-B*).

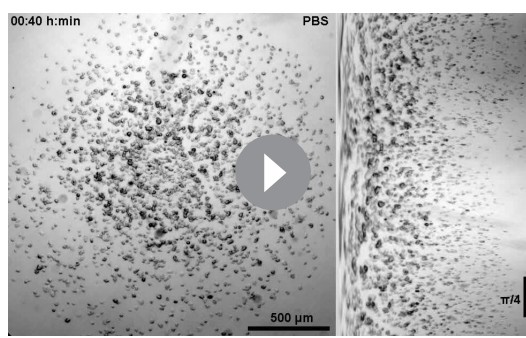

**Video 6.** Spot assay in phosphate buffer. Left: cell dynamics show the formation and migration of a ring of cells up to 4 hr at which point it started disintegrating and aggregates started forming around 10 hr. Right: polar visualization of cell dynamics with angles shown vertically and distance to the center horizontally. This visualization clearly shows the early propagation of a ring of cells.

https://elifesciences.org/articles/64731#video6

# A general framework for traveling waves in cells undergoing aerotaxis and division

From there onwards, we propose a mathematical framework that investigates general conditions under which collective behavior of cells undergoing cell division and aerotaxis is triggered. The aim was to confirm conclusions already obtained experimentally or through the cellular Potts model and to decipher the contribution of cell division to the collective behavior, while also keeping the framework relatively general such that it may be applied to other types of collective cellular behavior.

We first considered models of the form given by (*Equations 1 and 2*), independently of the exact shape of the advection term $a(C, \nabla C)$ or division term $r(C)$. Because of its relevance for the study of planar front propagation, we studied these models in a planar symmetry ($\rho = \rho(t,x), C = C(t,x)$) instead of a radial symme-

try ($\rho = \rho(t,r), C = C(t,r)$), neglecting thereby any curvature effects. We were interested in the study of a single front moving from left to right. Introducing the front speed $\sigma$, the front corresponds to a stationary solution in the moving frame $z = x - \sigma t$, that is, a traveling wave profile, satisfying:

$$-\sigma \frac{\partial \rho}{\partial z} = D \frac{\partial^2 \rho}{\partial z^2} - \frac{\partial}{\partial z}(a(C, \partial_z C)\rho) + r(C)\rho \quad (3)$$

In the general case, the theoretical analysis of such profiles and the determination of the front speeds $\sigma$ seem out of reach due to the coupling with the reaction-diffusion equation on the $O_2$ concentration. Nonetheless, it is possible to derive simple relations between the shape of the wave and the speed of propagation. By integrating (*Equation 3*) over the line, we obtain:

$$\begin{aligned}(\sigma - a(C(-\infty), \partial_z C(-\infty)))\rho(-\infty) \\ = \int r(C(z))\rho(z)dz\end{aligned} \quad (4)$$

This equation balances the net flux of cells to the far left-hand with the amount of mass created by heterogeneous (oxygen-dependent) cell division. We illustrated this relationship with the experimental data from *Figure 1E*. In order to approximate the term $\int r(C(z))\rho(z)dz$, we used an observation that we made through numerical simulations: cell division stops roughly at half of the peak, meaning that cells left to the peak do not divide, while cells right to the peak continue dividing (see *Figure 4E* and *Figure 6— figure supplement 1*). Therefore, we approximated by

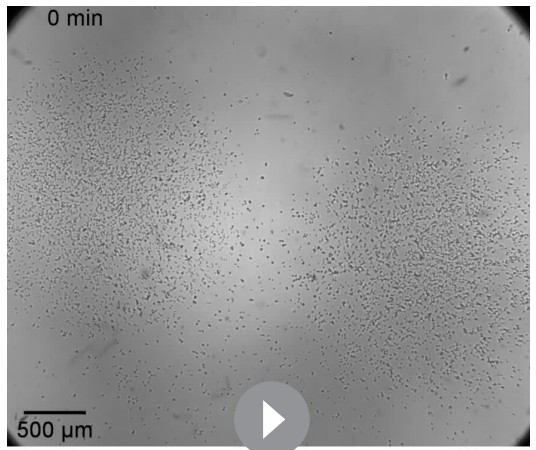

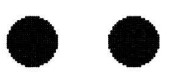

**Video 7.** Ring fusion during experiments (top) and as predicted by the Potts model (bottom). Note that this Potts model is a non-quantitative version and, as a result, space and time are in arbitrary units and thus not shown.

https://elifesciences.org/articles/64731#video7

a rectangle method $\int r(C(z))\rho(z)dz = \rho r_0 L/2$, where $L$ is the length spanned by the ring and $\rho$ is the average cell density in the ring. As there is supposedly no advection $a(C, \partial_z C)\rho = 0$ at $z = -\infty$ this yields the approximation $\sigma \approx \frac{r_0 L \rho}{2\rho(-\infty)}$. Quantitatively, we assume $L$ to be on the order of 300µm (**Figure 1E**) and $\frac{\rho}{\rho(-\infty)}$, the ratio between cell densities in the ring and in the bulk of cells, to be on the order of 4 (**Figure 1E**). This yields an estimate of the wave speed, based solely on the shape of the cell density profile, of $\sigma \approx 0.9$ µm/min.

## Mathematical analysis of the 'Go or Grow' hypothesis

The difficulty to study (**Equation 3**) analytically led us to propose a simpler version of the mean-field model that recapitulates the two key ingredients, cell division and aerotaxis, in an original way. Although it deviates from the reference Potts model in the details, it has the advantage of being analytically solvable. Cells have two distinct behaviors, depending on the O$_2$ concentration. Below a certain threshold $C_0$ cells move preferentially upward the oxygen gradient (go), with constant advection speed $a_0$, but they cannot divide. Above the same threshold they divide (grow) and move randomly without directional bias. This model may be considered as a strong simplification of (**Equation 1**), here restricted to the one-dimensional space, where:

$$a(C, \partial_x C) = a(C)\,sign(\partial_x C),\ \text{with}\ a(C) = \begin{cases} 0, & \text{if}\ C > C_0 \\ a_0, & \text{if}\ C < C_0 \end{cases} \text{and}\ r(C) = \begin{cases} r_0, & \text{if}\ C > C_0 \\ 0, & \text{if}\ C < C_0 \end{cases}. \tag{5}$$

The coupling between (**Equations 1 and 2**) then goes merely through the location of the oxygen threshold $C_0$. This elementary 'Go or Grow' model was meant to 1- demonstrate that its simple ingredients suffice to trigger a collective motion and 2- determine the relative contributions of cell division and aerotaxis on the speed of the ring in a general framework.

Interestingly enough, in this case (**Equation 3**) admits explicit traveling wave solutions (see more details in the Materials and methods section). Moreover, an explicit formula for the wave speed was obtained (**Figure 6C** and Materials and methods for a detailed derivation):

$$\sigma = \begin{cases} a_0 + \frac{r_0 D}{a_0}, & \text{if}\ a_0 \geq \sqrt{r_0 D} \\ 2\sqrt{r_0 D}, & \text{if}\ a_0 \leq \sqrt{r_0 D} \end{cases} \tag{6}$$

To the best of our knowledge, this analytical formula is new and captures basic features of a wave under a single self-generated gradient. It is remarkable that Formula (**Equation 6**) does not depend on the dynamics of oxygen consumption and diffusion. Furthermore, Formula (**Equation 6**) presents a dichotomy according to the relative size of aerotaxis strength $a_0$ and the quantity $\sqrt{r_0 D}$: in the case of small-bias (i.e. $a_0 \leq \sqrt{r_0 D}$), the wave speed $\sigma$ is independent of aerotaxis and coincides with the so-called Fisher's wave speed $2\sqrt{r_0 D}$. This speed is related to the Fisher-KPP equation (**Aronson and Weinberger, 1975**; **Fisher, 1937**; **Kolmogorov et al., 1937**), which describes front propagation under the combined effects of diffusion and growth (without advection). However, in the case of large-bias (i.e., $a_0 > \sqrt{r_0 D}$), aerotaxis is strong enough to contribute to the speed and the wave speed increases $\sigma > 2\sqrt{r_0 D}$.

Based on these observations, we propose the fraction $\varphi = 2\sqrt{r_0 D}/\sigma$ as a measure of the relative contribution of cell division and diffusion to the overall wave speed. Indeed, when aerotaxis is absent (or as in the small-bias case not contributing to the wave speed), the value of $\varphi$ is 1 and the wave is driven by cell division and unbiased random motion, that is, a reaction-diffusion wave. In the large-bias case, $1/\varphi$ describes how much faster the wave travels, compared to if it were only driven by diffusion and division. We illustrated the behavior of $\varphi$ with a heatmap (**Figure 6D**) as a function of the parameters $a_0$ and $\ln(2)/r_0$ (the doubling time of the cell population), the diffusion coefficient being fixed to its experimental value $D$=30 µm$^2$/min.

We confront this reasoning with the experimental data: as a rough approximation with $a_0 = 1$ µm/min in experiments, assuming a doubling time of 8 hr for Dd cells, $r_0 = ln2/480$ min$^{-1}$, we are clearly in the case of large bias ($\sqrt{r_0 D} = 0.2$ µm/min) and (**Equation 4**) yields $\sigma = 1.04$ µm/min while the fraction $\varphi = 40\%$. The wave travels 2.5 times faster than a wave merely driven by cell division, showing that in this case the dominant ingredient to set the wave speed is aerotaxis. Still, our results can similarly be applied to other systems in which this balance could be different. Finally, note that the

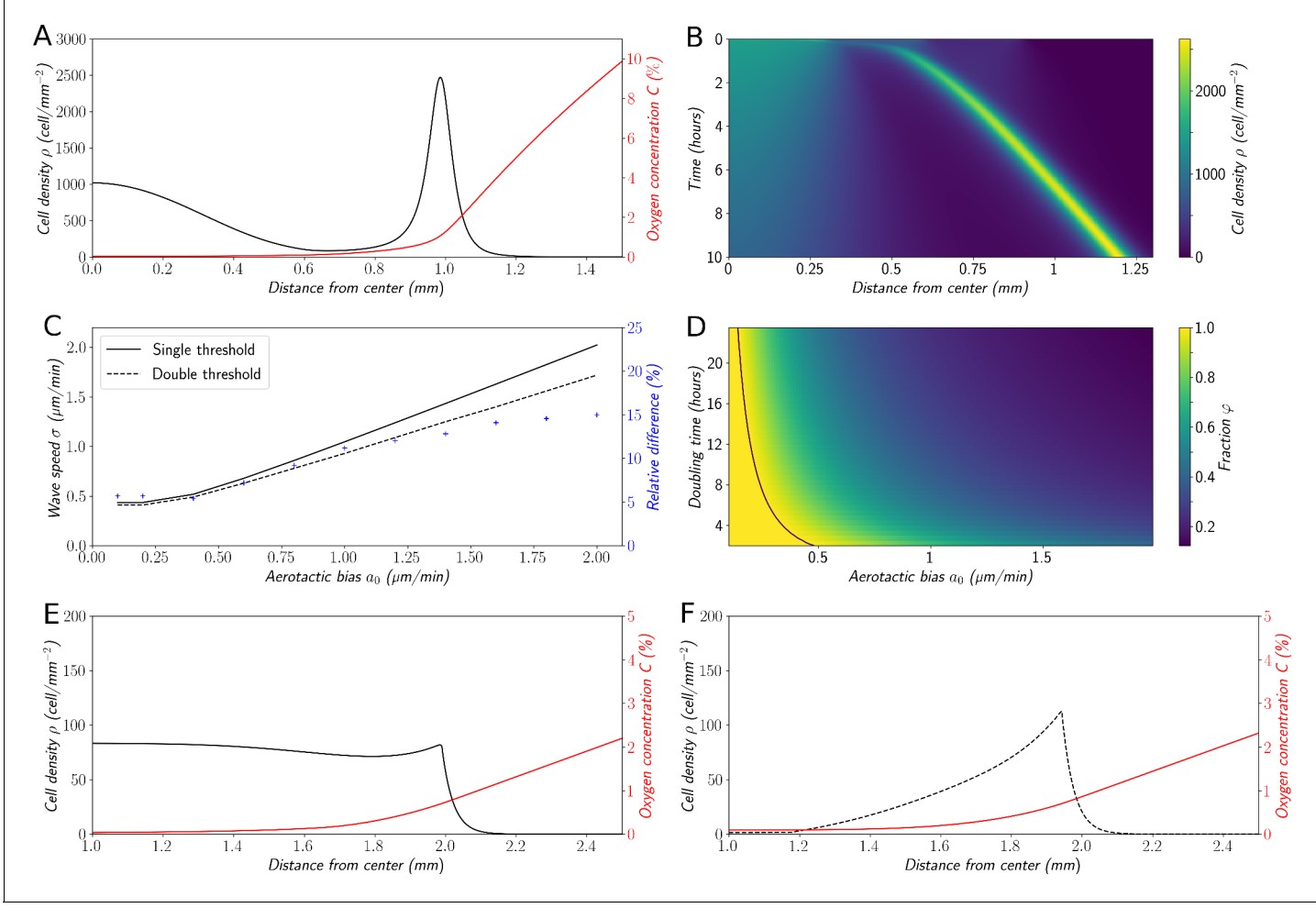

**Figure 6.** Variations on the 'Go or Grow' hypothesis. (**A**) Cell density and $O_2$ concentration profiles for the mean-field model (**Equations 1 and 2**). (**B**) Corresponding kymograph of cell density (colorbar in cells/mm$^2$) as a function of time and distance to the center. (**C**) Comparison of wave speeds for the elementary 'Go or Grow' model, given by Formula **Equation 6**, and the 'Go or Grow' model with a second threshold, obtained by numerical simulation (solid and dotted lines respectively). The relative difference between the speeds of the two models is represented by crosses. (**D**) Heatmap of $\varphi = 2\sqrt{r_0 D}/\sigma$ as a measure of the relative contribution of cell division to the overall wave speed $\sigma$ in the space parameter $\ln(2)/r_0$ and $a_0$ for the 'Go or Grow' model (**Equation 5**), where $\sigma$ is given by Formula **Equation 6**. The curve $a_0 = \sqrt{r_0 D}$ is depicted in black. (**E**) Cell density and $O_2$ concentration profiles for the 'Go or Grow' model with $a_0 = 1\,\mu m/min, r_0 = ln2/480\,min^{-1}$ and $C_0 = 0.7\%$. (**F**) Cell density and $O_2$ concentration profiles for the 'Go or Grow' model with two thresholds: cells undergo aerotaxis with a constant advection speed $a_0 = 1\mu m/min$ when the $O_2$ concentration is in the range $\left(C_0', C_0\right)$ with $C_0 = 0.7\%$, $C_0' = 0.1\%$. In both cases, thresholds coincide with the cusps in the profiles.

The online version of this article includes the following source data and figure supplement(s) for figure 6:

Source data 1. Zip file containing raw data for *Figure 6* and associated Python code for simulations.

Figure supplement 1. Structural variations of (*Equation 1*).

Figure supplement 1—source data 1. Zip file containing raw data for *Figure 6—figure supplement 1* and associated Python code for simulations.

density profile of the model (*Figure 6E*) does not present a sharp front peak as in experiments (*Figure 1D,E*), Potts simulations (*Figure 3B,C*) or in the mean field model (*Figure 6A,B*). We will show below that it can be slightly modified to change the profile of the fronts while keeping the analytical results relevant thus describing a whole class of systems (*Figure 6E* and *Figure 6—figure supplement 1*).

## Inside dynamics of the wave front

The wave speed of the elementary 'Go or Grow' model coincides with Fisher's speed, that is $\sigma = 2\sqrt{r_0 D}$, in the regime of small bias ($a_0 \leqslant \sqrt{r_0 D}$). This is the signature of a *pulled wave*,

meaning that the propagation is driven by the division and motion of cells at the edge of the front, with negligible contribution from the bulk, and little diversity in the expanding population. In contrast, when the bias is large ($a_0 > \sqrt{r_0 D}$) then the wave speed in (*Equation 6*) is greater than Fisher's speed. This is the signature of a *pushed wave*, meaning that there is a significant contribution from the bulk to the net propagation, with an expanding population maintaining diversity across expansion, see *Birzu et al., 2018*; *Stokes, 1976* for insights about the dichotomy between pulled and pushed waves. In particular, it was conjectured that the ratio $\varphi = 2\sqrt{r_0 D}/\sigma$ proposed above controls the transitions between different regimes of diversity loss in a wide class of reaction-diffusion models (*Birzu et al., 2018*; *Birzu et al., 2019*).

In order to explore this dichotomy between pulled and pushed waves, we used the framework of neutral labeling (*Roques et al., 2012*) in the context of PDE models. We colored fractions of the density profile during wave propagation to mimic labeling of cells with two colors. Then, we followed numerically the dynamics of these fractions, and quantified the mixing of the two colors. Our results were in perfect agreement with (*Roques et al., 2012*), extending their results beyond classical reaction-diffusion equations to equations which also include advection (see Materials and methods). In the case of large bias (*Figure 7A–C*), the wave is pushed and the profile is a perfect mixture of blue and yellow cells at long times. Contrarily, the wave is pulled in the regime of small bias: only cells that were already initially in the front, here colored in blue (*Figure 7B–D*), are conserved in the front, whilst yellow cells at the back cannot catch up with the front.

In the absence of associated experimental data, we explored the cellular Potts model with such neutral labeling. The results were in agreement with the PDE simulations (*Figure 7—figure supplement 1*) showing a clear, rapid mixing of the two cell populations under the propagation of a pushed wave in the regime of experimental parameter values.

## Robustness of the conclusions to structural variations of the model

We voluntarily defined our elementary 'Go or Grow model' as a rough simplification of our original mean-field model in order to keep it solvable and extract a general formula for the front speed and an analysis of the relative contribution of diffusion/division and aerotaxis in that respect. However, many experimental systems will not conform to the hypothesis underlying this model (in particular the shapes of the aerotactic response and cell division modulation). In order to investigate whether the conclusions drawn from the elementary 'Go or Grow' model extend to more general situations, we decided to submit it to structural variations and check if the results obtained above still held. First, we made the hypothesis of a second oxygen threshold $C_0' < C_0$, below which cells are not sensitive to gradients any longer (*Figure 6F*). In the general case, we were not able to do a thorough analysis of this model, but through numerical exploration we found that the propagation speed remained close to the value given by formula (*Equation 6*) (at most 15% of relative difference in a relevant range of parameters, *Figure 6C*). Intuitively, the main contribution to the collective speed is the strong bias inside the high-density area at intermediate levels of $O_2$, whereas cells at levels below the second threshold $C_0'$, where the dynamics of both models diverge, do not contribute much to the collective speed. We also noticed that cell density profiles (*Figure 6F*) were much closer to experimental observations and results obtained through the cellular Potts model or the original mean-field approach. Moreover, the wave speed is no longer independent of the oxygen dynamics. In the Materials and methods section, we pushed further the analysis with a specific form of oxygen consumption and developed a specific case of such a 'Go or Grow' model with a second threshold, where we were able to conduct its complete analysis. *Figure 7—figure supplement 2* shows that the conclusions concerning the contribution of growth to the wave speed are robust. Finally, we show on this modified 'Go or Grow' model that our conclusions regarding how the behavior can switch from a pulled to a pushed wave remain true as well (*Figure 7—figure supplement 3*) demonstrating that our results can be generalized to a variety of different systems showing the propagation of a front in response to a single self-generated gradient.

To go beyond this first variation with two oxygen thresholds, we also investigated the influence of the shape of the aerotactic response such as linear or logarithmic gradient sensitivity. *Figure 6—figure supplement 1* shows the qualitative outcomes of these different models. This numerical exploration indicates that a wide combination of the two key ingredients, aerotaxis and cell division, can drive the propagation of a stable wave with various density profiles. Cell division at the edge

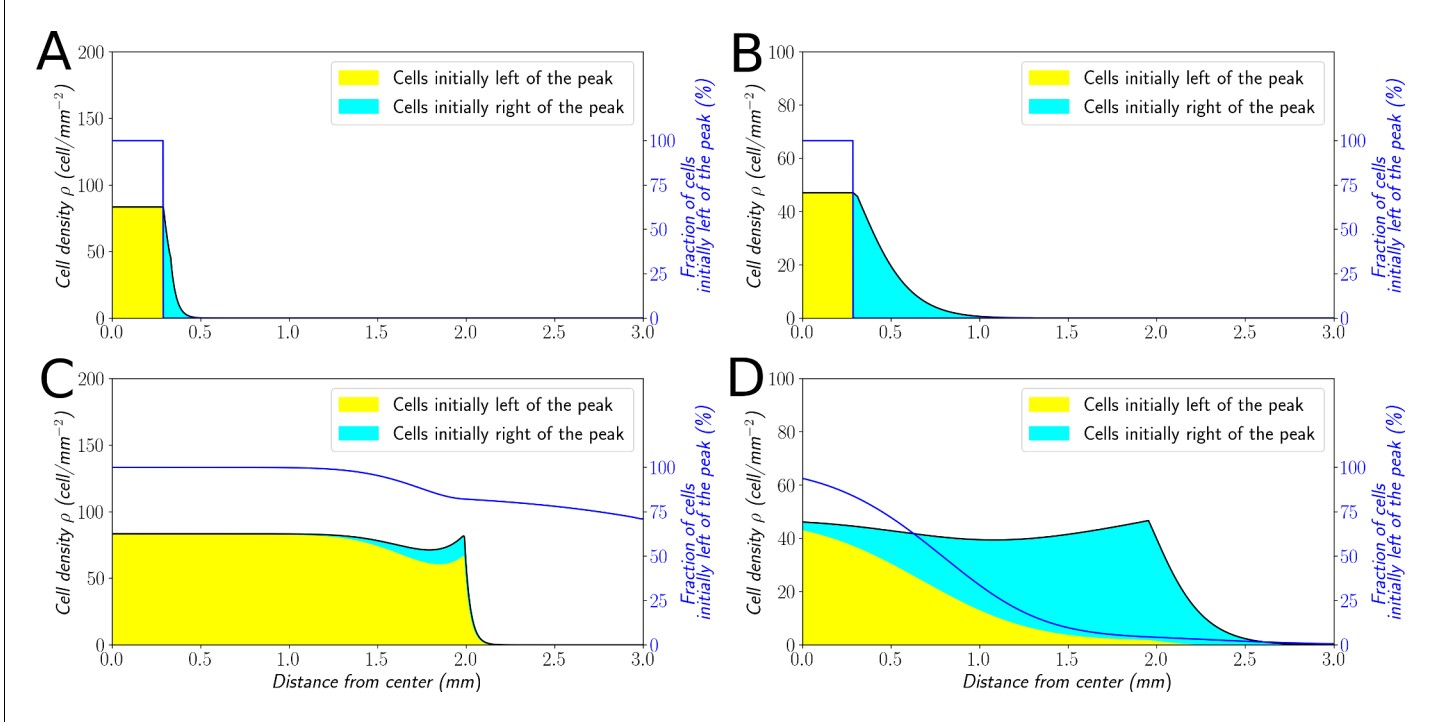

**Figure 7.** Classification of the expansion type in the 'Go or Grow' model. Cells initially on the left-hand side or right-hand side of the peak get labeled differently (A and B). The labeling is neutral and does not change the dynamics of the cells. We let evolve the two colored population for some time and observe the mixing of the colors (C and D). (A and C) With $a_0 = 1m \cdot min^{-1}$, the wave is pushed wave and after some time the front undergoes a spatially uniform mixing. (B and D) With $a_0 = 0.1 \mu m \cdot min^{-1}$, the wave is pulled and only the fraction initially in the front is conserved in the front. $r_0 = ln2/480min^{-1}$ and $C_0 = 0.7\%$ for all conditions.

The online version of this article includes the following source data and figure supplement(s) for figure 7:

**Source data 1.** Zip file containing raw data for *Figure 7* and associated Python code for simulations.

**Figure supplement 1.** Mixing in Potts models.

**Figure supplement 1—source data 1.** Raw data for *Figure 7—figure supplement 1*.

**Figure supplement 2.** Heatmap of $\varphi = 2\sqrt{r_0 D}/\sigma$ as a measure of the relative contribution of cell division to the overall wave speed $\sigma$ in the space parameter $1/r_0$ and $a_0$ for the 'Go or Grow' model with a second threshold, under the specific condition that $O_2$ consumption term be $b(C) = b_0$ and that $O_2$ concentration may be negative (see Materials and methods).

**Figure supplement 2—source data 1.** Zip file containing raw data for *Figure 7—figure supplement 2* and associated Python code for simulations.

**Figure supplement 3.** Classification of the expansion type in the 'Go or Grow' model with a second-threshold.

**Figure supplement 3—source data 1.** Zip file containing raw data for *Figure 7—figure supplement 3* and associated Python code for simulations.

yields a net flux of cells, backward in the moving frame, that sustains the wave propagation in the long term, but may have a relatively small contribution to the wave speed.

## Discussion

Overall, our results demonstrate the ability of Dd cells to respond to hypoxia through both aerotactic and aerokinetic responses. Both of these behaviors could be very important to help Dd cells to navigate complex hypoxic environments they encounter in the soil. In addition, our results are a confirmation of the ability of self-generated gradients to serve as very robust, long-lasting directional cues in environmental navigation, a property which has recently emerged in a variety of systems (*Cremer et al., 2019*; *Tweedy and Insall, 2020*). Finally, our work goes beyond theses results as it demonstrates that oxygen can play the role of the attractant in self-generated gradients therefore potentially extending the physiological relevance of the use of such cues in collective migration.

In addition, although our experimental results were obtained on simple, 2d experiments, our findings can generalize to more complex cases. The fact that the dense front of cells follows a constant oxygen concentration (*Figure 3E*, *Figure 4E*) provides a hint that any situation in which cell density

is locally high enough to trigger hypoxic conditions will also lead to a similar behavior. Then, depending on the dimensionality of the system, its architecture and the position of possible oxygen sources, we hypothesize that a similar front will develop and follow isoconcentration lines. Indeed, the original experiments of Adler (*Adler, 1966*) and more recent developments (*Cremer et al., 2019*; *Fu et al., 2018*; *Saragosti et al., 2011*) on bacteria demonstrated that similar ingredients as the ones presented here can lead to front propagation in both 1d and 2d situations. Similarly, using an under agarose assay, it was demonstrated that self-generated gradients of degraded folate induce a group migration of cells in bands (in 1D) or rings (in 2D spots) up to 4 mm (*Tweedy and Insall, 2020*). Beyond the dimensionality of the system, it was also shown that self-generated gradients allow cells to solve mazes by locally degrading an attractant that has a source at the exit of the maze (*Tweedy et al., 2020*). Our results are in total agreement with these past examples. To further show the generality of the underlying principles, we ran some 3D Potts simulations using a qualitative version of our model. Briefly, we show that in three dimensions, if oxygen is provided on all sides, a spherical front of cells starts moving outwards (*Video 8*). However, if we assumed that the bottom of the space was completely deprived of oxygen (i.e. a symmetry breaking situation that can be encountered in various physiological situations), this front was migrating upwards only in a half-spherical shape (*Video 8*). Our 2d results can therefore be extended to any other situations and they show that the key to proper steering are high enough cell densities and the creation of robust self-generated gradients.

While aerotaxis is well established for bacteria, its role is often invoked in multicellular organisms to explain various processes in development or cancer progression but very few *in vitro* studies were conducted to prove it is an efficient and operating mechanism or to understand the molecular mechanisms at play during aerotaxis. Deygas et al. showed that confined epithelial colonies may trigger a self-generated $O_2$ gradient and an aerotactic indirect response through a secondary ROS self-generated gradient (*Deygas et al., 2018*). Gilkes et al. showed that hypoxia enhances MDA-MB231 breast cancer cell motility through an increased activity of HIFs (*Gilkes et al., 2014*). HIFs activate transcription of the Rho family member RHOA and Rho kinase 1 (ROCK1) genes, leading to cytoskeletal changes, focal adhesion formation and actomyosin contractions that underlie the invasive cancer cell phenotype. This study suggests a role for aerotaxis in tumor escape, but it only demonstrates aerokinesis as $O_2$ gradients were not imposed to probe a directed migration toward $O_2$. Using a microfluidic device, the same cancer cell line was submitted to various oxygen levels as well as oxygen gradients (*Koens et al., 2020*) but the observed aerotactic response was not clear.

By contrast, the experimental results presented here with *Dd* show a strong response to hypoxia. Within 15 min, cells exhibit an aerokinetic and aerotactic response when exposed to externally imposed $O_2$ gradients (*Figure 2*). Self-generated $O_2$ gradients are produced within 20 min (*Figure 3* and *Figure 1—figure supplement 2*). But this cellular response is within the equilibration time of the oxygen distribution (*Figure 2—figure supplement 6*). Hence we can consider the cellular response as almost instantaneous with *Dd*. The difference with previously studied cells is probably due to the extreme plasticity of the rapidly moving amoeboid cells (*Dd*) and their almost adhesion independent migration mechanism (*Friedl et al., 2001*) while mesenchymal cancer cells move slower by coordinating cytoskeleton forces and focal adhesion (*Palecek et al., 1997*).

The quick response of *Dd* in directed migration assays has been largely exploited to decipher the molecular mechanisms at play during chemotaxis (*Nakajima et al., 2014*). The molecular mechanisms used for $O_2$ sensing and its transduction into cellular response are for the

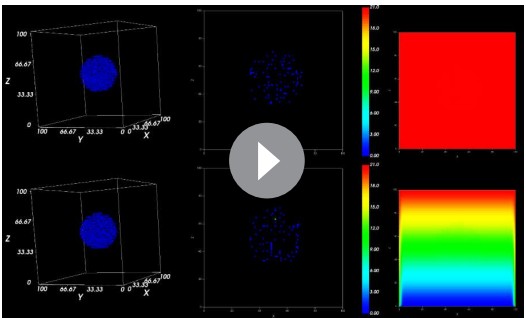

**Video 8.** 3D Potts simulations. Top : with oxygen sources on all sides. Bottom: with oxygen sources on all sides except on the bottom. The left column shows the behavior of the whole cell assembly in 3D (original cells are in blue, cells created during the process are shown in green). The middle column is an xz slice of the cell behavior to show that the 3d structures are indeed spheres or pseudo-spheres. The right column is the same slice as in the middle but showing oxygen profiles as a colormap.

https://elifesciences.org/articles/64731#video8

moment unknown but we can expect that the $O_2$ molecular sensors modulate cytoskeleton organization, particularly localized actin polymerization/depolymerization through some of the molecular components involved in classical chemotaxis toward folate or cAMP (*Pan et al., 2016*; *van Haastert et al., 2017*). However, new and unexpected mechanisms cannot be excluded.

The finding that migrating cells can influence the direction of their own migration by building chemoattractant gradients is not new. Several species of bacteria can move preferentially toward oxygen or nutrient as reported by *Adler, 1966*. However, this mechanism was only recently reported in eukaryotes (*Stuelten, 2017*): melanoma cells that break down lysophosphatidic acid (LPA) and generate a LPA gradient in the vicinity of the tumor (*Muinonen-Martin et al., 2014*), *Dd* colonies that generate folate gradients (*Tweedy et al., 2016*) or for the migration of the zebrafish lateral line primordium through a self-generated chemokine gradient (*Donà et al., 2013*; *Venkiteswaran et al., 2013*). The dispersal of melanoma cells is particularly instructive. The stroma surrounding the tumor acts as a source of LPA. The tumor cells act as a sink for LPA. As long as LPA is present in the environment a steady wave of migrating melanoma cells propagates away from the initial tumor over long distances and long time periods.

The self-generated LPA (melanoma) and folate (*Dd*) gradients were modeled with a simple numerical model that was able to predict the steady wave. In particular, it predicted an invasive front where cells are exposed to a steep chemoattractant gradient, followed by a 'trailing end' where the gradient is shallow and fewer cells migrate with poor directionality (*Tweedy and Insall, 2020*). It also predicted that the wave may have a less marked front, and/or a smaller speed, or even vanishes if the cell density was too low due to insufficient chemoattractant removal. All these features are surprisingly similar to our experimental measurements of cell density and $O_2$ profiles (*Figure 1E*, *Figure 3C*). The atmospheric $O_2$ that diffuses through the culture medium and eventually the plastic surfaces is the chemoattractant. The $O_2$ consumption triggers hypoxia that in turn generate an aerotactic response toward $O_2$ in a very narrow range of $O_2$ concentrations (0–1.5%) (*Figure 3C*). The exact value of the lower $O_2$ threshold value will deserve future investigations. The exact nature of the cellular response at these extremely low $O_2$ levels, and in a very shallow gradient, also has yet to be clarified.

Our different models unveil a set of basic assumptions which are sufficient for collective motion of cells without cell-cell interactions (attractive or otherwise), in contrast with (*Sandonà et al., 1995*). Cell growth is necessary to produce a long-standing wave without any damping effect. However, it may not be the main contribution in the wave speed, depending on the relative ratio between directional motion (the bias $a_0$), and reaction-diffusion (the Fisher half-speed $\sqrt{r_0 D}$). In the case where the former is greater than the latter, the wave is due to the combination of growth and directional motion and it is pushed. This result differs particularly from the Fisher-KPP equation with constant advection (meaning with uniform migration and division) where the wave speed is $a_0 + 2\sqrt{r_0 D}$ and the wave is pulled. In the experiments under study, we estimate directional motion to contribute the most to the cell speed, ruling out the possibility of seeing a pulled wave driven by cell division and diffusion at the edge of the front.

In conclusion, we demonstrate the remarkable stability of collective motion driven by self-generated gradients through depletion of oxygen. Through coupled dynamics, these gradients give rise to long lasting, communication-free migrations of entire colonies of cells which are important both from ecological and developmental points of view. In the case presented here where oxygen plays the role of the depleted attractant, this could prove to be a very general mechanism as oxygen is ubiquitous and always consumed by cells.

## Materials and methods

### Cell preparation

The AX2 cell line was used and cultured in HL5 media (Formedium, Norfolk, UK) at 22°C with shaking at 180 rpm for oxygenation (*Sussman, 1987*). Exponentially growing cells were harvested, counted to adjust cell density to the desired one, typically 2000 cells/μL.

## Observations and analysis of self-generated aerotaxis by cell confinement (spot assay)

For the spot assay, 1 µL of cell suspension containing 1000–8000 cells (typically 2000) was carefully deposited on a dry surface using a 1 µL syringe (Hamilton, Reno, NV, USA). The dry surface was either the nunclon treated surface of Nunc six wells polystyrene plates for usual experiments (ThermoFisher Scientific, Waltham, MA, USA) or polydimethylsiloxane surface (PDMS, Sylgard 184, Dow Corning, Midland, MI, USA) for experiments on oxygen-sensing films. The drop was incubated for 5 to 7 min in humid atmosphere at 22℃ before gently adding 2 ml of HL5 medium without detaching the spotted cells forming a micro-colony. A 14 mm or 18 mm diameter round glass coverslip cleaned in ethanol, thoroughly rinsed in HL5 was kept wet and deposited on top of it. In some experiments, fluorescein FITC at 16 µM was added to the HL5 medium and confocal slices were taken, showing that confined *Dictyostelium* cells were not compressed by the coverglass but separated from it by a layer of medium of about 50 µm (*Figure 1—figure supplement 1*).

The outward spreading of the *Dictyostelium* micro-colony was observed at 22℃ in transmission with three types of microscope: (i) a TE2000-E inverted microscope (Nikon, Tokyo, Japan) equipped with motorized stage, a 4x Plan Fluor objective lens (Nikon) and a Zyla camera (Andor, Belfast, Northern Ireland) using brightfield for most of the experiments lasting 16 hr (*Figure 1A*), (ii) a binocular MZ16 (Leica, Wetzlar, Germany) equipped with a *TL3000 Ergo* transmitted light base (Leica) operated in the one-sided darkfield illumination mode and a LC/DMC camera (Leica) for experiments over days (*Figure 1B*) and finally (iii) a confocal microscope (Leica SP5) with a 10x objective lens for a few larger magnification experiments (*Figure 1C*).

For computing densities, cell positions were determined using the built-in Find Maxima plugin in ImageJ (National Institutes of Health, Bethesda, MD, USA) through a custom made routine. Data analysis and plotting was performed in Matlab (Mathworks, Natick, MA, USA). For density profiles (*Figure 1E*) and kymographs (*Figure 1D*), the center of the colony was defined as the center of mass of all cells detected at all times. Cell positions were then turned into radial coordinates and cells were counted within concentric crown regions. Densities were calculated by dividing this count by the area of each crown.

Density profiles such as the ones showed in *Figure 1E* were treated to automatically extract the position, width and density of a ring in various experiments and at various time points. Density profiles were first stripped of values lower than 500 cell/mm$^2$ in order to avoid asymmetric baselines behind and in front of the ring. Resulting profiles were then fitted in Matlab by a Gaussian function with a non-zero baseline. The non-zero baseline corresponds to density in the bulk, the maximum of the Gaussian gives ring position, its height added to the non-zero baseline gives the cell density in the ring and its width the width of the ring.

## Cell tracking, diffusion coefficients, and aerotactic biases

After retrieving cells' positions with optimized ImageJ macros based on Find Maxima, the individual trajectories were reconstructed with a squared-displacement minimization algorithm (http://site.physics.georgetown.edu/matlab/). Data were analysed using in-house Matlab programs. Timelaspse microscopy experiments devoted to cell tracking in the spot assay experiments was acquired at a high frame rate (1 frame every 15 s) (*Figure 1—figure supplements 4–5*) due to the very high cell density in the ring region (up to 2000 cells/mm$^2$, *Figures 1E* and *3C*). For the microfluidic experiments, as cells were plated at a lower density (less than 200 cells/mm$^2$), 1 min time intervals was used to track cell trajectories (*Figure 2C*). In order to highlight aerotactic biases, cells displacements over various time lags dt (dt up to 60 min) were projected in the radial direction for spot assays and in the gradient direction X for microfluidic experiments and eventually divided by dt to obtain velocity biases. Individual biases were then averaged within bins of equal distance (*Figure 2D*, *Figure 1—figure supplement 5*, *Figure 4—figure supplement 1*). Individual effective cell diffusion constants were measured as the square of their displacement over their entire trajectory divided by the trajectory time length and divided by 4. These measurements were then similarly averaged over bins (*Figure 1—figure supplement 4*).

## Microfluidic-based oxygen gradient generator: design, fabrication, and cell injection

A schematic of the present double-layer microfluidic device is shown in *Figure 2A*. It is made of several layers of PDMS mounted on a bottom glass coverslip. The overall diameter $D$ of the microfluidic device is 27 mm and the overall thickness $H$ is 4 mm. Three parallel media channels are positioned for cell culture, and two gas channels are positioned at a height $H_g$=0.5 mm above the media channels to allow gas exchange between the channels during cell culture. The horizontal distance between the two gas channels narrows step-by-step (2 mm, 1 mm, and 0.5 mm gaps, respectively), thus yielding to generate different gradients of oxygen concentration along the media channels simultaneously. All channels are 125 µm high and 2 mm wide, and therefore, the media and gas channels are separated by a PDMS wall of 375 µm thickness. A polycarbonate (PC) film (26 mm in diameter and 0.5 mm thickness) is embedded inside the device at a height $H_f$=1 mm from the bottom coverslip to prevent oxygen diffusion from the atmosphere. The cartesian coordinate origin was set at the center of the media channel (median axis), and the $x$ and $y$-directions were defined as parallel to media and gas channels respectively (*Figure 2A*). The $z$-direction was set to the vertical direction from the top of the bottom coverslip.

The manufacturing steps are as follow. The media channel and gas channels were drawn with AutoCAD (Autodesk, Mill Valley, CA, USA) and replicated in SU-8 photoresist using classical photolithography techniques. These SU8 molds were silanized to make it non-adherent and reusable. PDMS was mixed at a 10:1 ratio of base:curing agent, poured over each mold to a thickness of $H_g$, and cured in an oven at 60°C for more than four hours. On top of the cured PDMS layer of the gas channels, the above-mentioned PC film with 3 mm port holes punched at the locations of the media and gas channel ports was positioned. Additional PDMS was then poured over the PC film until the total PDMS layer became 3.5 mm thick, then the PDMS layer was cured in an oven at 60°C overnight. The PDMS layers of the media and gas channel patterns were peeled off the silicon wafers and cut into 27 mm diameter circles. The PDMS layer with the gas channel pattern was punched to form inlets and outlets 2 mm in diameter to allow the infusion of gas mixtures. The channel-patterned surface of the PDMS layer with the gas channels and the top surface of the other PDMS layer with the media channels were plasma treated (PDC-001-HP; Harrick Plasma Inc, Ithaca, NY, USA) to bond with each other. After incubating the bonded PDMS mold overnight in an oven at 60°C, 2 mm diameter inlets were punched to allow access to the media channels, respectively. Finally, the channel-patterned side of the PDMS mold and a 35 mm-diameter glass bottom dish with or without covered by an oxygen sensing film were plasma treated and bonded each other.

Measurements of aerokinesis in homogenous environments were performed using a homemade glass-duralumin environmental chamber to perform random motility assays (*d'Alessandro et al., 2018*; *Golé et al., 2011*).

*Dictyostelium* cells were seeded in the media channels at density of $2\times10^6$ cells/ml, and the cell culture medium was filled in the glass bottom dish up to the height covering the PDMS mold. Cells were allowed to adhere to the bottom surface (bare glass or coverglass covered with a sensing film) for 15 min.

## Gas control and injection

We used a controlled oxygen concentration for three types of experiments: (i) to calibrate oxygen sensing films (see below), (ii) to create the oxygen gradients within microfluidic devices (see below) or (iii) to insure a pure hypoxic condition (pure $N_2$) at the end of the spot assay experiment. The gas mixture (0% to 21% $O_2$ in $N_2$) was prepared in a gas mixer (Oko-lab 2GF-MIXER to mix compressed AIR with 100% N2 or HORIBA STEC MU-3405, Kyoto, Japan to mix pure O2 and N2) by mixing pure $O_2$ (or air) and pure $N_2$. Free sensing films for calibration (i) or for the spot assay (iii) were placed inside 6-wells plates and the multiwells were placed in an environmental chamber fitting our microscope stage (H301-K-frame, Okolab, Pozzuoli, Italia). Gas was injected at about 500 mL/min in this chamber. Eventually, multi-wells were drilled to a diameter of 25 mm and the sensing films were glued with a silicone adhesive on the plate bottom to reduce the background noise from fluorescence. For microfluidic experiments, the tubes from the mixer were connected to the gas channels and gas was injected at a controlled flowrate (between 60 and 180 mL/min) into the device.

## Oxygen-sensing film preparation

Oxygen-sensing films were prepared by inserting the luminescent $O_2$-sensitive dye 5,10,15,20-Tetrakis-(2,3,4,5,6-pentafluorophenyl)-porphyrin-Pt(II) (PtTFPP, Por-Lab, Porphyrin-Laboratories, Scharbeutz, Germany) in a 4:1 PDMS:curing agent thin layer spin-coated on 30 mm to 35 mm rounded coverglasses. Briefly, 17 mg of PtTFPP was dissolved in 5 mL chloroform and thoroughly mixed with 2.8 mg of PDMS and 0.7 mg curing agent. The mixture was degassed in a vacuum chamber for 5 hr. About 0.5 mL to 1 mL of this solution was spread on the coverglass and spin-coated for 2 min at 500 rpm with a final speed of 2000 rpm during 10 s to flatten the edge bead. Chloroform was allowed to evaporate overnight while the polymer cured at 60°C. The final PtTFPP sensor film had a dye concentration of 4 mmol/L and was 25 µm thick. This thickness was measured using a ContourGT-K 3D Optical microscope (Bruker, Billerica, MA, USA) after removing a piece of film with a surgical blade. Sensing films were stored in dark. They were used to measure the oxygen concentration in self-generated $O_2$ gradients (spot assay) and for microfluidic experiments with controlled $O_2$ gradients.

## Fluorescence microscopy for oxygen measurements

Fluorescence images of $O_2$-sensing films (either for film calibration or for in situ oxygen measurements in the spot assay or in microfluidic devices) were taken with two inverted epifluorescence microscopes: (i) a TE2000-E inverted microscope (Nikon) equipped with motorized stage, a 4x Plan Fluor objective lens (Nikon), a X-Cite Series 120PC illumination lamp, a TRITC bandpass filter cube and a Zyla camera (Andor) (*Figure 3Aii*, *Figure 2—figure supplement 3*), (ii) a IX83 inverted microscope (Olympus, Tokyo, Japan) equipped with a motorized stage, a UPlanSApo 4x objective lens (Olympus), a U-HGLGPS lamp (Olympus), a RFP bandpass filter and a Zyla camera (Andor). This second microscope was used for mosaic imaging, in order to scan the whole dimension of the three media channels (about 1 cm in length) thanks to the dedicated imaging software cellSens (Olympus) (*Figure 2—figure supplement 1*).

## Oxygen-sensing film calibration

Calibration was carried out with the sensing films in air, in water or in HL5 culture medium. We applied gas concentration ramps with steps of 5 min for calibration in air (time to exchange fully the gas composition of the chamber and tubes, as $O_2$ almost instantaneously diffuses within the 25 µm thick sensing film) and with much longer steps (i.e. 2–4 hr) for calibrations in liquid. There is indeed an additional diffusion time in the PDMS intermediate layer of our microfluidic devices or in the medium height of a Petri dish: typically, a few minutes for a 0.5-mm-thick PDMS layer and 1h30 for a 2.7-mm-thick liquid layer in a dish.

Timelapse fluorescence images we recorded and signal intensity $I$ was measured in ROIs of typically 64x64 pixels in various positions of the image, especially along a line scanning the middle of the image (*Figure 2—figure supplement 1B*, *Figure 2—figure supplement 3A*). The response of the sensing film in the presence of oxygen can be modeled by a linear Stern-Volmer relationship:

$$\frac{I_0 - B_g}{I(C) - B_g} = 1 + KC$$

where C is the oxygen concentration expressed as a percentage of oxygen in the injected gas phase (nearly 21% for atmospheric conditions), $I_0$ is the reference intensity in the absence of oxygen, $B_g$ is the background intensity independent from the oxygen sensitive signal of the PtTFPP molecules and $K$ is the Stern-Volmer constant used as an indicator of the sensing film sensitivity.

Notice that the background is usually not included in the Stern-Volmer relationship but a representative background image ($O_2$ independent) is subtracted prior to intensity measurements (*Nock et al., 2008*; *Thomas et al., 2009*). This $O_2$-independent background value can be deduced from the fluorescence of a plain PDMS film prepared in the same conditions than the sensing film but devoid of PtTFPP molecules (i.e. a standard) (*Thomas et al., 2009*). We tested that procedure that is basically working but we choose to include the background as a fitting parameter because illumination conditions may change between the sample and the standard (especially the focus plane that affects the focused height of autofluorescent medium above the surface). The slight changes in thickness and PtTFPP composition of sensing films at the large spatial scales we are interested here (3–6 mm wide images, *Figure 2—figure supplement 1A–D*, *Figure 2—figure supplement 3A–C*)

are another source of heterogeneity especially for the $K$ value. For those reasons, we apply the Stern-Volmer relation with $K$ and $B_g$ as a fitting parameters in many different small regions of interest (ROI) of the surface. For each ROI, we found that the measured intensities follow perfectly the Stern–Volmer relation (i.e. $C$ linearly increases with the Stern-Volmer parameter $(I_0-B_g)/(I(C)-B_g)-1$, *Figure 2—figure supplement 2B*) and that $K$ and $B_g$ are clearly uncorrelated, in particular $B_g$ depends on the illumination pattern but not $K$.

The illumination pattern is clearly visible on fluorescence images at 21% $O_2$. For instance, the large field of view of *Figure 2—figure supplement 1B* reconstituted by the multi-area module of the microscope displays an up and down landscape in the 21% intensity (*Figure 2—figure supplement 2C*) due illumination changes in the periphery of each overlapped area but also due to the slightly different fluorescence in the gap region of the microfluidic device and especially at its interfaces. The single large field of view of *Figure 2—figure supplement 3A* taken with our second microscope setup displays a dome shaped pattern with 20% intensity difference between the center and borders (*Figure 2—figure supplement 3D*). The background value $B_g$ is very correlated with the 21% signal variations in both imaging configurations (*Figure 2—figure supplement 2C*, *Figure 2—figure supplement 3D*) and we can for each experiment calibrate a linear relationship between between $B_g$ and $I(21\%)$ (*Figure 2—figure supplement 2D*, *Figure 2—figure supplement 3F*).

As already stated, the background is due to the autofluorescence of the various media (PDMS layer and coverglass for the sensing film, bottom plastic dish if any, and surrounding fluid). For *Figure 2—figure supplement 1*, the microfluidic device was filled with pure water and for *Figure 2—figure supplement 3*, the calibration was performed in the autofluorescent HL5 medium before spotting the cells. Sometimes the sensing film was placed in a non-drilled well and in that case the strong autofluorescence of the plastic bottom of the plate becomes a major source of background (not shown). Finally, $B_g$ also includes the read noise $RN$ of the camera which is a constant independent of the light output or exposure time. For *Figure 2—figure supplement 2*, we measured $RN=108$ A.U. and hence a light background $B_g^* = B_g-RN=15$ A.U. which is half the 'true oxygen dependent signal' at 21% $O_2$, $I(21\%)-RN=30$ A.U. The maximum deviation of $B_g$ from the linear fit in *Figure 2—figure supplement 5D* is about 1.5 A.U. Hence a relative error $1.5/15=10\%$ for $B_g^*$ will be taken in the following. For *Figure 2—figure supplement 3*, we measured $RN=100$ A.U. and at the top of the bell curve, $B_g^*=1400-100=1300$ A.U. while $I(21\%)^*=1600-100=1500$ A.U. (*Figure 2—figure supplement 3D*). Hence, the background is nearly 87% of the signal due to the HL5 autofluorescence. Nevertheless, the maximum relative deviation from the linear fit is smaller at about $25/1300\approx2\%$ of $B_g^*$. All these values will be used for the error analysis of the oxygen profiles below.

For uncovered (free) sensing films the sensibility $K$ ranges between 3 and 5 $\%^{-1}$ and is very constant, weakly dependent on the illumination pattern (*Figure 2—figure supplement 2D* and blue points in *Figure 2—figure supplement 3E*). When films are covered with a coverglass, the fluorescence under hypoxic conditions (0%) increases significantly on the covered region (*Figure 2—figure supplement 3C*) but not at 21% (*Figure 2—figure supplement 3B*). As a results, K, which is proportional to this ratio, increases significantly (*Figure 2—figure supplement 3E*) but not $B_g$ and $I(21\%)$ (*Figure 2—figure supplement 3D*) confirming that $K$ and $B_g$ are independent. This increase in $K$ is probably due to a local temperature increase: the coverglass adsorbs more heat from the microscope illumination light and this heat is difficult to evacuate due to the confinement. In principle, for the spot assay experiment, it would be necessary to perform an independent calibration with the Stern-Volmer relation in the covered situation. However, this is difficult due to the very long time required to equilibrate the oxygen level under the confinement far from the coverglass boundary (this is why we started the Stern-Volmer fit at $ROI_7$ in *Figure 2—figure supplement 3B,D,E*). A too long procedure causes other problems such as medium evaporation, stage or focus drift... To avoid that, we decided to apply the protocol described in the image analysis pipeline of *Figure 2—figure supplement 4*. First, we perform a gas calibration ramp and do a Stern-Volmer analysis in various points of an uncovered sensing film (the subscript $U$ is for uncovered) where $I_{0U}$ is reliable in order to get the linear background relation $B_{gU}=\alpha I(21\%)_U +\beta$ and to measure the measure the ratio $R=I_{0U}/I(21\%)_U$. Reliable means here any point if gas mixture is applied uniformly when calibrating in a dish or just underneath the gas channel in microfluidic devices. Second, we choose the reference fluorescence image $I(21\%)$ immediately before starting any experiment (i.e. just after covering the spot, or just before applying the gradient in the gas channels). From that image, we build a $B_g$ image as $\alpha I$

*(21%)+β* (as $B_g$ is the same for uncovered and covered case) and eventually we build a reconstituted $I_0$ image as $R\,I(21\%)$. Finally, we subtract and divide images with the 'Image calculator' of ImageJ (i.e. pixel by pixel) following the Stern-Volmer model and hence get a $K$-value image map and subsequently an oxygen map (***Figure 2—figure supplement 4C–D***). This enables to correct non-homogeneous illumination conditions or non-homogeneous sensing film properties as well to quickly estimate error bars on the oxygen map from the estimated errors on $B_g$, $I(21\%)$ and $I_0$ detailed below.

We already discussed the error on the background. In principle, *I(21%)* is a reference image (hence error free), however as an experiment (especially the spot assay) may run overnight we need somewhere to evaluate drift in the absolute intensity for instance by running an overnight timelapse experiment with a sensing film under ambient gas conditions. This error is added on *I(21%)* and was estimated as 2% of the true fluorescence signal corrected from read noise *I(21%)-RN*. Error on $I_0$ could be much larger. As the intensity *I(C)* is strongly nonlinearly increasing with the oxygen concentration *C*, if we measure an intensity $I_0^*$ corresponding to a residual small oxygen level $C_0^*$, we need to correct the true $I_0$ value using the relation $I_0 - B_g \approx I_0^* \approx I_0\left(1 + KC_0\right)$. Due to oxygen leakage along tubes and within our environmental chamber, when applying 100% $N_2$, we measured a residual $C_0 \approx 0.15$ $O_2$ in a culture medium dish using a bare fiber oxygen sensor coupled to its commercial oxymeter (Firesting, Pyroscience, Aachen, Germany). Hence with a typical $K$=5 value, we obtain a very large discrepancy between the measured fluorescence $I_0^*$ and the ideal one: $I_0 \approx 1.75 I_0$, but finally this discrepancy is not really dramatic on the measured error again due to the non-linearity.

The effect of these different error sources on the measured oxygen map is presented in ***Figure 2—figure supplement 1E*** and ***Figure 2—figure supplement 3H*** for a typical microfluidic and spot experiments. Even if we make a 1.75 error on $I_0$, this has little effect on the profiles except in the very hypoxic region when C<0.25% where the error exceed 50%. But even in the region around C=1%, the error is less than 10%. The error on *I(21%)* on the other hand has a significant effect on the high oxygen regions but less on the hypoxic regions. Finally, the background error is relatively visible in the intermediate oxygen concentration region (very visible on the side of the spot in ***Figure 2—figure supplement 3H***, but also to some extend around the median axis at C~10% in the microfluidic experiment, ***Figure 2—figure supplement 1E***). Finally, we defined error bars with *(max-min)/2* values of the calculated *C* when exploring the estimated errors discussed above. In the 0.5–1.5% region were we observe most of the interesting aerotactic behaviors with *Dictyostelium* cells, the precision on the oxygen concentration $\Delta C/C$ is less than 0.3. For the purpose of this paper, we can conclude that aerotaxis and aerokinesis occurs undoubtedly between C=0% and C=2% (***Figure 2***).

Numerical simulation of oxygen tension. Oxygen tension inside the device was computed using commercial finite element software (COMSOL Multiphysics 5.5; COMSOL, Inc, Burlington, MA, USA). The gas flow in the individual channels were simulated by solving the Navier-Stokes equations coupled with mass continuity for an incompressible fluid:

$$\rho_G(\boldsymbol{u} \cdot \nabla)\boldsymbol{u} = \mu_G \Delta \boldsymbol{u} - \nabla \boldsymbol{p},$$
$$\rho_G \nabla \cdot \boldsymbol{u} = 0,$$

where u is the velocity vector, *p* is the pressure, and $\rho_G$ and $\mu_G$ are the gas density and viscosity (taken as 1 kg/m$^3$ and $10^{-5}$ Pa.s, respectively). The spatial and temporal distribution of oxygen inside the device was then calculated by solving the convection-diffusion equation:

$$\frac{\partial \boldsymbol{c}}{\partial \boldsymbol{t}} = \boldsymbol{Dc} - \boldsymbol{u} \cdot \nabla \boldsymbol{c},$$

where *c* is the oxygen concentration, *D* is the diffusion coefficient of oxygen, and *T* is the time.

The device was assumed to be in an atmosphere containing 21% $O_2$. Medium at 21% $O_2$ concentration was supplied to media channels. Gases containing 0% and 21% $O_2$ were respectively supplied to the left-hand and right-hand side gas channels at 30 ml/min to generate an oxygen gradient. Zero pressure and convection flux conditions were set at the outlets of the gas channels, and a no-slip condition was applied on the channel walls for fluid flow analysis. Boundary conditions for oxygen concentration were set according to Henry's law. Oxygen concentration at the interfaces between the PDMS and gas phase (atmosphere and gas mixture in the gas channel) was set

correspondingly to the product of the solubility coefficient of oxygen in PDMS and the partial pressure of oxygen. At the interfaces between PDMS and media or gel, a partition condition was applied, which balanced the mass flux of oxygen to satisfy continuity of partial pressure of oxygen:

$$\frac{c_{PDMS}}{S_{PDMS}} = \frac{c_{channel}}{S_{channel}},$$

where $c_{PDMS}$ and $S_{PDMS}$ are the oxygen concentration and the solubility of oxygen in the PDMS, respectively, and $c_{channel}$ and $S_{channel}$ are those in the media and gel channels. Moreover, oxygen consumption by cells was considered by setting an outward flux of oxygen of $6 \times 10^{-8}$ [mole/(m$^2$.s)] on the bottom of the media channels (calculated as $b\rho^*$ where $b=1.2.10^{-16}$ mole/(cell.s) is the oxygen molar consumption per *Dd* cell per unit of time (*Torija et al., 2006*) and $\rho^*=500$ cell/mm$^2$ is the highest density used in the device).

The diffusion constants of oxygen in the various media were taken to be $2.10^{-9}$, $4.1.10^{-9}$ and $2.10^{-12}$ m$^2$/s for culture medium, PDMS and PC, respectively. Oxygen solubility at 1atm were taken to be 219 (close to the measured value, see below), 1666 and 1666 μM for culture medium, PDMS and PC, respectively (PDMS and the PC values were assumed to be the same since they are reportedly within the same range *Merkel et al., 2000*; *Moon et al., 2009*). The computational models consisted of approximately 1,135,000 computational elements. The initial condition of oxygen concentration in each material was set to 21% O$_2$ everywhere (219 μM).

## Potts models

Potts model simulations were run using CompuCell3D (*Swat et al., 2012*) with a mix of prebuilt modules and home-made Python steppables in particular to implement the modulation of aerotactic strength by local oxygen levels. Most parameters were fitted to experimental measurements and both time and length scales were also adapted to achieve quantitative simulations. In all simulations, we used Compucell's Volume module which applies to all cells a Hamiltonian of the form:

$$H_{volume} = \lambda_v (V - V_{cell})$$

where $V$ is the volume of a cell and $V_{cell}$ a target volume set to 2 pixels. This already set the length scale of our simulations to 1pixel = 10μm. $\lambda_v$ was set to 800. These values were adapted to reproduce the cell speeds observed in the microfluidic experiments. To achieve this relationship, we also decided to fix that one step of the simulation (Monte Carlo Step) was meant to represent 0.1s.

Aerotaxis was modeled using CompuCell's built-in chemotaxis plugin. This leads to a new term in the Hamiltonian of the form

$$H_{chemotaxis} = \lambda_{aero} \Delta C$$

where $\Delta C$ is the difference in oxygen concentration $C$ between the source and target pixels of a flip and $\lambda_{aero}$ is the aerotactic strength. Key to our model is thus the fact that we made $\lambda_{aero}$ different for each cell and dependent on the local oxygen concentration. This modulation was fitted to the microfluidic experiments and set, in the general model as:

$$\lambda_{aero}(C) = \frac{800}{1 + e^{\frac{C-0.7}{0.2}}}$$

where C is the oxygen concentration at the center of mass of a cell. *Figure 5* shows variations on that relationship which are:

$$\lambda_{aero}(C) = \frac{1225}{1 + e^{\frac{C-0.7}{1.5}}}$$

Based on experimental observations, we also made the effective temperature of the model different for each cell and dependent on local oxygen concentrations. This allowed us to reproduce the aerokinetic effect and the modulation of the temperature was fitted to reproduce the cell diffusion constants measured in the microfluidics experiment. The main model thus uses the following relationship for temperature T

$$T(C) = 85 + \frac{105}{1 + e^{\frac{C-0.7}{1}}}$$

*Figure 5* and *Figure 5—figure supplement 1* show variations on this relationship, which is simply replaced by a constant value of T (115, 135 or 155).

Another key aspect of the models is the oxygen field, which is implemented using CompuCell's DiffusionSolverFE module. In the case of the microfluidics experiments, the oxygen field was made to be constant (no diffusion, no consumption by cells) and fitted on experimental measurements of this gradient shown in *Figure 2B*. The oxygen concentrations were expressed in % giving 0 and 21 as natural boundaries. The actual oxygen profile varied in the x direction only as:

$$C(x) = \frac{21.38}{1 + e^{0.031*(x-200)}}$$

where x is the position in pixels in the simulation which was run on a 400 by 800 *xy* grid.

For spot assays, oxygen was allowed to diffuse freely. In our time and length units, the diffusion constant of oxygen in liquids ($2.10^3 \ \mu m^2.s^{-1}$) is two pixel$^2$ step$^{-1}$.

In terms of consumption, we took the oxygen consumption by Dicty cells to be $1.2.10^{-16}$ mole/(cell.s) (*Torija et al., 2006*) and we measured the oxygen solubility in HL5 medium as 250 µM (measurement with a bare fiber sensor plugged to a a Firesting oximeter, Pyroscience, Germany). Taking the measured vertical confinement of 50 µm (*Figure 1—figure supplement 1*), the amount of oxygen available, at maximum, above a single pixel of the simulation is $1.25.10^{-15}$ moles, which we define, in arbitrary units, to be 21. We can then turn the consumption of a single cell into a consumption per pixel given that the typical size of a cell is two pixels and per time step, each representing 0.1 s. We end up with a consumption, in our arbitrary units of 0.1 pixel$^{-1}$ step$^{-1}$ which is only applied to pixels occupied by a cell. Of note, in case an occupied pixel had a remaining oxygen level of less than this values, then consumption was set at this oxygen level so that all oxygen was consumed. The last ingredient in oxygen dynamics is the leak of oxygen coming from the bottom of the multiwall plate. Assuming complete hypoxia on the cells' side, this would lead to a net flux of oxygen of *DC/e* where D is the oxygen diffusion constant in polystyrene, C is the oxygen concentration on the outside and e is the thickness of the polystyrene bottom. This leads to a flux by unit surface, in our Potts units of 0.001 pixel$^{-1}$.step$^{-1}$. We therefore implement a source of oxygen for all pixels in the simulation, whether they are occupied by a cell or not, of the form:

$$secretion(C) = 0.001 \frac{21-C}{C}$$

where C is the local oxygen concentration at the considered pixel.

This was sufficient to faithfully reproduce the formation time of the rings. Finally, the spot simulations were run on a 500 by 500 pixels grid and we imposed boundary conditions to the oxygen field as a constant concentration of 21, the borders acting as a source of oxygen just like the edges of the coverslip in the experiments.

Cell division was also set to experimental observations. Given a doubling time of 8 hr, we implemented random divisions at each time point, each cell having a 1/ (8h * 3600 s/h * 10step/s) = $3.10^{-6}$ chance of dividing. However, cell division was turned off at low oxygen concentrations (<0.7%). In *Figure 5*, a simulation is shown were this probability was set to 0 for all cells in all conditions.

In terms of initial conditions, the microfluidic simulations were started from a homogenous cell density, each cell being initialized on a grid: two pixels per cells and a six pixel gap to the next neighbor in all directions. For the spot simulations, cells were seeded in three circular, concentric regions of decreasing density. The first region was set to be 30 pixels (300 µm) in radius with a gap of 1 pixel between each cell, the second one spanned the radii between 30 and 60 pixels with a gap of two pixels between each cell and the last one spanned between 60 and 90 pixels with a gap of 3 pixels. This lead to an initial colony with a radius of 900 µm and between 1900 and 2000 cells, both very similar to experiments.

## Mean-field model, Go or Grow model and simulations

Both diffusion equations (*Equations 1 and 2*) were discretized through a time-backward space-centered difference scheme with an upwind discretization for the advection operator. In the case of the mean-field model, we were considering (*Equations 1 and 2*) in a radial symmetry, which lead to the following discretization for $\rho$:

$$\frac{\rho_i^{n+\frac{1}{2}}-\rho_i^{n-\frac{1}{2}}}{\Delta t}-D\frac{\left(R_i+\frac{\Delta x}{2}\right)\left(\rho_{i+1}^{n+\frac{1}{2}}-\rho_i^{n+\frac{1}{2}}\right)-\left(R_i-\frac{\Delta x}{2}\right)\left(\rho_i^{n+\frac{1}{2}}-\rho_{i-1}^{n+\frac{1}{2}}\right)}{R_i\Delta x^2}$$
$$+\frac{1}{R_i\Delta x}\cdot\left\{\begin{array}{l}cR_ia_i^n\rho_i^{n+\frac{1}{2}}-R_{i-1}a_{i-1}^n\rho_{i-1}^{n+\frac{1}{2}}\ if\ C_i^n\geq C_{i-1}^n\\R_ia_i^n\rho_i^{n+\frac{1}{2}}-R_{i+1}a_{i+1}^n\rho_{i+1}^{n+\frac{1}{2}}\ if\ C_i^n<C_{i-1}^n\end{array}\right\}=r_i^n\rho_i^{n+\frac{1}{2}}$$

, where $a_i^n=\lambda\left(C_i^n\right)\frac{C_i^n-C_{i-1}^n}{\Delta x}, r_i^n=r\left(C_i^n\right)$ and $R_i$ the distance from the center.

In the case of the Go or grow model with its planar symmetry, the discretization for $\rho$ was:

$$\frac{\rho_i^{n+\frac{1}{2}}-\rho_i^{n-\frac{1}{2}}}{\Delta t}-D\frac{\rho_{i-1}^{n+\frac{1}{2}}-2\rho_i^{n+\frac{1}{2}}+\rho_{i+1}^{n+\frac{1}{2}}}{\Delta x^2}+\frac{1}{\Delta x}\cdot\left\{\begin{array}{l}a_i^n\rho_i^{n+\frac{1}{2}}-a_{i-1}^n\rho_{i-1}^{n+\frac{1}{2}}\ if\ C_i^n\geq C_{i-1}^n\\a_i^n\rho_i^{n+\frac{1}{2}}-a_{i+1}^n\rho_{i+1}^{n+\frac{1}{2}}\ if\ C_i^n<C_{i-1}^n\end{array}\right\}=r_i^n\rho_i^{n+\frac{1}{2}},\ \text{with}\ a_i^n=a\left(C_i^n\right).$$

Concerning the equation on Oxygen concentration *Equation 2*, the consumption term $-b(C)\rho$ was expressed in two different manners: either $b(C)=b_0$ and a non-negativity constraint was added on the Oxygen concentration, just as it was the case in the cellular Potts model, or $b(C)=min\left(b_0,b_0\frac{C}{C_0'}\right)$ which leads to an oxygen consumption that goes to zero in the region of very low concentration $C<C_0'$ and therefore ensures non-negativity for $C$ under a sufficiently small time step $\Delta t$. Both expressions led to qualitatively similar results, but we opted for the latter in all the simulations presented here. Finally, the discretization scheme for $C$ in the planar symmetry was:

$$\frac{C_i^{n+1}-C_i^n}{\Delta t}-D_{oxy}\frac{C_{i-1}^n-2C_i^n+C_{i+1}^n}{\Delta x^2}=-b_i^n\rho_i^{n+\frac{1}{2}},\ \text{with}\ b_i^n=b\left(C_i^n\right).\ \text{For the radial symmetry:}$$

$$\frac{C_i^{n+1}-C_i^n}{\Delta t}-D\frac{\left(R_i+\frac{\Delta x}{2}\right)\left(C_{i+1}^{n+1}-C_i^{n+1}\right)-\left(R_i-\frac{\Delta x}{2}\right)\left(C_i^{n+1}-C_{i-1}^{n+1}\right)}{R_i\Delta x^2}=-b_i^n\rho_i^{n+\frac{1}{2}}$$

The schemes were coded in Python language. All simulations of (*Equations 1 and 2*) shown in this article were carried out with a mesh size $\Delta t=0.02min$ and $\Delta x=1\mu m$. The values used for the constants are: $D=30\mu m^2\cdot min^{-1}$ (effective cellular diffusion constant), $C_0=0.7\%O_2$ (threshold for cell division), $C_0'=0.1\%O_2$ (lower threshold in the two threshold 'Go or Grow' model, below which cells stop aerotaxis), $D_{oxy}=1.2\cdot10^5\mu m^2\cdot min^{-1}$ (oxygen diffusion constant in medium), $r_0=ln2/480min^{-1}$ (rate of cell division) and $b_0=0.01\%O_2min^{-1}cell^{-1}$ (using the equivalence $1.25.10^{-15}$ moles = 21 %O_2 discussed above in Potts model section).

The scheme on $C$ was supplemented with the boundary condition $C(L)=21\%O_2$. We have chosen $L=9mm$ for the Go or grow model to match experimental conditions. For the mean-field model, we have chosen $L=2.5mm$ in order to match the cellular Potts model for which size was a concern for computation time.

In the mean-field model, the initial condition for $\rho$ was taken the same as in the cellular Potts model. For the other simulations, initial conditions for $\rho$ and $C$ were chosen such that they were already close to the expected stationary profile.

We measured the speed of the wave $\sigma$, once the wave profile was qualitatively stable, by considering the evolution of the point $\bar{x}(t)$ such that $C(t,\bar{x}(t))=C_0$.

## Mathematical analysis of the 'Go or Grow' model

We present below a preliminary analysis of the 'Go or Grow' model. A more detailed mathematical investigation of this model will be carried out in a separate article.

1.The 'Go or Grow' model admits explicit traveling wave solutions.

We recall that $z=x-\sigma t$ is the spatial variable in the moving frame at (unknown) speed $\sigma>0$. We seek a pair of stationary profiles, resp. the density $\rho(z)$ and the oxygen concentration $C(z)$. We assume that $C(z)$ is an increasing function. By translation invariance, we set without loss of generality that $C(0)=C_0$, so that *Equation 3* becomes:

$$\begin{cases} -\sigma\frac{\partial\rho}{\partial z} = D\frac{\partial^2\rho}{\partial z^2} - a_0\frac{\partial\rho}{\partial z}, \ if \ z<0 \\ -\sigma\frac{\partial\rho}{\partial z} = D\frac{\partial^2\rho}{\partial z^2} + r_0\rho, \ if \ z>0 \end{cases} \tag{7}$$

Furthermore, the function $\rho(z)$ must satisfy at $z=0$ the following relation (i.e. the continuity of the flux) :

$$\sigma\frac{\partial\rho}{\partial z}(0^+) - \sigma\frac{\partial\rho}{\partial z}(0^-) = -\frac{a_0}{D}\rho(0) \tag{8}$$

Thus the equation becomes a second order differential equation with piecewise constant coefficients on each half-line, that can be solved explicitly.

For $z<0$, the solution is of the form $A + Be^{\frac{a_0-\sigma}{D}z}$. From **Equation 4** we observe that $\sigma \geq a_0$ equivalently $\frac{a_0-\sigma}{D} \leq 0$ and as $\rho$ is bounded, it implies that $B=0$.

For $z>0$, we look at the roots of the characteristic polynomial $P(\mu) = D\mu^2 + \sigma\mu + r_0$. We note that to yield a nonnegative solution, we need $\sigma^2 \geq 4r_0D$.

If $\sigma = 2\sqrt{r_0D}$, then the solution is of the form $(Cz+D)e^{-\sqrt{\frac{r_0}{D}}z}$ and with relation **Equation 8**, we obtain $\rho(z) = A\left(\frac{\sqrt{Dr_0}-a_0}{D}z+1\right)e^{-\sqrt{\frac{r_0}{D}}z}$ and observe that in this case, we necessarily have $a_0 \leq \sqrt{r_0D}$.

If $\sigma>2\sqrt{r_0D}$, the solution is of the form $A'e^{-z} + B'e^{\mu_+z}$, with $\pm = \frac{-\sigma\pm\sqrt{\sigma^2-4r_0D}}{2D}$. By arguments exposed in **Van Saarloos, 2003**, solutions with initial datum localized cannot decrease exponentially at a rate $\mu > -\sqrt{\frac{r_0}{D}}$, where $-\sqrt{\frac{r_0}{D}}$ corresponds to the exponential decay parameter when $\sigma = 2\sqrt{r_0D}$. This leads to $B' = 0$, as $\mu_+ > -\sqrt{\frac{r_0}{D}}$. Then $A = A'$, but in order to satisfy the $C^1$-discontinuity jump relation **Equation 7**, it must be that :

$$\mu_- = \frac{-a_0}{D} \tag{9}$$

**Equation 9** can be solved algebraically for $\sigma$, which yields $\sigma = a_0 + \frac{Dr_0}{a_0}$. Furthermore, we can rewrite **Equation 9** as follows $(2a_0 - \sigma) = \sqrt{\sigma^2 - 4r_0D}$, multiplying by $2a_0 + \sigma$, we find that $(4a_0^2 - \sigma^2) = (2a_0 + \sigma)\sqrt{\sigma^2 - 4r_0D}>0$, which leads to $a_0^2>\frac{\sigma^2}{4}>r_0D$.

Thus, we have disclosed all possible profiles. In the case $a_0 \leq \sqrt{r_0D}$ the profile travels at speed $\sigma = 2\sqrt{r_0D}$, whilst for $a_0>\sqrt{r_0D}$ the profile travels at speed $\sigma = a_0 + \frac{r_0D}{a_0}$.

One needs to verify that each of these profiles admits an associated oxygen profile that satisfies the condition $C(0) = C_0$, but the preceding profiles were defined up to the multiplicative constant $A$, by linearity of **Equation 7**. The differential equation on $C$ becomes with $\tilde{\rho}$ the solution given above for $A = 1$:

$$-\sigma\frac{\partial C}{\partial z} = D_{oxy}\frac{\partial^2}{\partial z^2} - b\left(A\tilde{\rho}\right)C \tag{10}$$

One concludes by checking that by monotonicity there exists a unique constant $A$ such that the solution to the differential **Equation 10** equation satisfies $C(0) = C_0$.

2. The wave is pushed in the case $a_0>\sqrt{r_0D}$.

A neutral fraction $v^k$ is defined as satisfying the following linear equation in the moving frame $z = x - \sigma t$:

$$\frac{\partial v^k}{\partial t} + Lv^k := \frac{\partial v^k}{\partial t} - \sigma\frac{\partial v^k}{\partial z} - D\frac{\partial^2 v^k}{\partial z^2} + \frac{\partial}{\partial z}\left(a(z)v^k\right) - r(z)v^k = 0, \tag{11}$$

with $v^k(0,z) = v_0^k(z)$ where we identify $a(z) = a(C(z))$ and $r(z) = r(C(z))$ for the sake of clarity. This corresponds biologically to staining the cells given by the initial distribution $v_0^k$ at time $t=0$ with a neutral label (**Roques et al., 2012**).

Defining $U(z) := \frac{(\sigma-a(z))}{D}z$, then we note that $Lf = -D\frac{\partial}{\partial z}\left(e^{-U}\frac{\partial}{\partial z}(e^U f)\right) - r(z)f$. This leads to setting $w := e^{\frac{U}{2}}v^k$ that satisfies the parabolic equation $\frac{\partial w}{\partial t} + \tilde{L}w = 0,$

with $\tilde{L}g := -De^{\frac{U}{2}}\frac{\partial}{\partial z}\left(e^{-U}\frac{\partial}{\partial z}\left(e^{\frac{U}{2}}g\right)\right) - r(z)g = -D\frac{\partial^2 g}{\partial z^2} + \left(\frac{U'^2}{4} - r(z) - \frac{a_0}{2}\delta\right)g$. The operator $\tilde{L}$ is self-adjoint in $L^2(R, dz)$ on the appropriate domain. Then by setting $\gamma := min\left(inf\left\{\frac{U'^2}{4} - r(z)\right\}, \left(\frac{r_0 D}{a_0}\right)^2\right) > 0$, one can first show that every element of the spectrum of $\lambda \in \sigma(\tilde{L})$ such that $\lambda < \gamma$ is an eigenvalue of $\tilde{L}$. Second, one shows that the only eigenvalue $\lambda$ of $\tilde{L}$ such that $\lambda < \gamma$ is $\lambda = 0$. Finally by standard theory of self-adjoint operators and semi-group theory, one obtains that $w(t) = Pw_0 + e^{-t\tilde{L}}(I - P)w_0$, where $\|e^{-t\tilde{L}}(I - P)w_0\|_{L^2(R,dz)} \leqslant e^{-\gamma t}\|w_0\|_{L^2(R,dz)}$. Translating these properties onto the neutral fraction $v^k$, we have that $v^k(t) \to \frac{\langle v_0^k, \rho\rangle_{L^2(R, e^U dz)}}{\langle \rho, \rho\rangle_{L^2(R, e^U dz)}}\rho$ at an exponential rate, where $\rho$ is the traveling wave profile calculated in the previous section. Therefore, each fraction converges to a fixed proportion of the whole population. We conclude that after some time the wave becomes a perfect mix of each neutral fraction. This corresponds to the definition of a pushed wave according to *Roques et al., 2012*.

3. The wave is pulled in the case $a_0 \leq \sqrt{r_0 D}$.

The preceding reasoning does not apply to this case and the intuition is clear, as the wave speed coincides with Fisher's $\sigma = 2\sqrt{r_0 D}$, which is typically the signature of a pulled reaction- diffusion front. In order to prove the pulled nature of the front, we consider $w^k = \frac{v^k}{\rho}$, where $\rho$ is the corresponding wave profile. By computation, $w^k$ then satisfies the following PDE:

$$\frac{\partial w^k}{\partial t} - \beta(z)\frac{\partial w^k}{\partial z} - D\frac{\partial^2 w^k}{\partial z^2} = 0 \text{ with } \beta(z) = \begin{cases} \frac{2\sqrt{r_0 D} - a_0}{D}, & if z < 0 \\ 2\frac{\sqrt{r_0 D} - a_0}{(\sqrt{r_0 D} - a_0)z + 1}, & if z \geq 0 \end{cases} \quad \text{and set } \eta \text{ a positive solution to the}$$

differential equation $\eta' = \beta\eta$. As $\beta(z) \geq 0$ and $\beta'$ bounded above, it can be shown by arguments similar to *Roques et al., 2012*, that under the integrability condition $\int \left(w_0^k(z)\right)^2\eta(z)dz < \infty$, the neutral fraction goes extinct, that is $\lim_{t=+\infty} \|(w^k)^2\eta\|_\infty = 0$, which characterizes a pulled wave in the framework of neutral fractions.

## Mathematical analysis of a specific 'Go or Grow' model with a second threshold

We present quickly a specific case for a 'Go or Grow' model with a second threshold, that is completely analytically solvable. We consider the advection term of the form $a(C)sign(\partial_x C)$ with $a(C) = \begin{cases} a_0, & if \ C_0' < C < C_0 \\ 0, & otherwise \end{cases}$, the division rate $r(C) = \begin{cases} r_0, & if \ C > C_0 \\ 0, & if \ C < C_0 \end{cases}$ and the O₂ consumption rate per cell $b(C) = b_0$, without including the constraint that the O₂ concentration $C$ be non-negative. Although this hypothesis seems physically non relevant, it is consistent with the fact that cells are not sensitive to O₂ concentration gradients below the threshold $C_0'$.

Given a traveling wave profile $\rho, C$ and the corresponding front speed $\sigma$, we suppose $C(0) = C_0$ and we introduce the spatial gap $h > 0$ between the two thresholds, i.e. $C(-h) = C_0'$, so that (*Equation 1*) becomes:

$$\begin{cases} -\sigma\frac{\partial\rho}{\partial z} = D\frac{\partial^2\rho}{\partial z^2}, & if \ z < -h \\ -\sigma\frac{\partial\rho}{\partial z} = D\frac{\partial^2\rho}{\partial z^2} - a_0\frac{\partial\rho}{\partial z}, & if \ -h < z < 0 \\ -\sigma\frac{\partial\rho}{\partial z} = D\frac{\partial^2\rho}{\partial z^2} + r_0\rho, & if \ z > 0 \end{cases} \tag{12}$$

Introducing a multiplicative constant $A$, $\rho$ is then of the shape:

$$\begin{cases} \rho(z) = A\left(B + Ee^{\frac{\sigma - a_0}{D}h}\right), & if \ z < -h \\ \rho(z) = A\left(B + Ee^{\frac{-(\sigma - a_0)}{D}z}\right), & if \ -h < z < 0 \\ \rho(z) = Ae^{-\mu z}, & if \ z > 0 \end{cases} \tag{13}$$

With $B = \frac{\sigma - D}{\sigma - a_0}$, $E = \frac{D - a_0}{\sigma - a_0}$ and $= \frac{\sigma + \sqrt{\sigma^2 - 4Dr_0}}{2D}$. We obtain the following condition, that establishes a one-to-one correspondence between $\sigma$ and $h$:

$$e^{\frac{\sigma-a_0}{D}h} = \frac{a_0(\sigma - \mu D)}{\sigma(a_0 - \mu D)} \tag{14}$$

The equation on $C$ becomes:

$$-\sigma\frac{\partial C}{\partial z} = D_{oxy}\frac{\partial^2}{\partial z^2} - b_0\rho \tag{15}$$

With the assumption that that $C$ be continuously differentiable, we can solve *Equation 15* for $C$:

$$\begin{cases} C(z) = FAz + G, \ if \ z < -h \\ C(z) = HAz + IAe^{\frac{-(\sigma-a_0)}{D}z} + J + Ke^{\frac{-\sigma}{D_{oxy}}z}, \ if \ -h < z < 0 \\ C(z) = LAe^{-\mu z} + Me^{\frac{-\sigma}{D_{oxy}}z} + C_{init}, \ if \ z > 0 \end{cases}$$

With $F = \frac{b_0}{\sigma}\left(B + Ee^{\frac{\sigma-a_0}{D}h}\right)$, $\quad G = C_0' + FAh$, $\quad H = \frac{b_0 B}{\sigma}$, $\quad I = \frac{b_0 D^2 E}{(\sigma-a_0)(D\sigma - D_{oxy}(\sigma-a_0))}$, $\quad L = \frac{b_0}{(D_{oxy}-\sigma)}$, $M = C_0 - LA - C_{init}$, $\quad\quad\quad\quad J = C_0 - IA - K \quad\quad\quad\quad$ and, $\quad\quad\quad\quad\quad\quad$ by setting $\quad \Delta = \frac{\sigma}{D}\left(H - F - I\left(\frac{\sigma-a_0}{D}\right)e^{\frac{\sigma-a_0}{D}h}\right) - \left(H + L\left(-\frac{\sigma}{D_{oxy}}\right) - I\left(\frac{\sigma-a_0}{D}\right)\right)\left(\frac{\sigma}{D_{oxy}}e^{\frac{\sigma}{D_{oxy}}h}\right)$, we have that $K = \frac{1}{\Delta}\left(H - F - I\left(\frac{\sigma-a_0}{D}\right)e^{\frac{\sigma-a_0}{D}h}\right)\frac{\sigma}{D_{oxy}}\left(C_{init} - C_0'\right)$ and $A = \frac{1}{\Delta}\left(\frac{\sigma}{D_{oxy}}e^{\frac{\sigma}{D_{oxy}}h}\right)\frac{\sigma}{D_{oxy}}\left(C_{init} - C_0'\right)$. This closes the system, but one more constraint remains, which is:

$$C_0' = -HAh + IAe^{\frac{(\sigma-a_0)}{D}h} + J + Ke^{\frac{\sigma}{D_{oxy}}h} \tag{16}$$

The front speed $\sigma$ of a traveling wave must therefore satisfy the implicit *Equation 16*. Finding a closed form for the solutions of *Equation 16* seems out of reach. Nevertheless, we can approximate the roots numerically, especially by noticing through numerical observation that *Equation 16* is monotone on the interval $\left(2\sqrt{r_0 D}, a_0 + \frac{r_0 D}{a_0}\right)$, where the root $\sigma$ is located. Hence through a dichotomy search algorithm we can find the speed $\sigma$ of the traveling wave with arbitrary accuracy.

## Acknowledgements

We thank R Fulcrand for his expert help in microfabrication; G Torch, C Zoude, G Simon and A Pied-noir for technical assistance. This study was supported by the CNRS - Mission pour les Initiatives Transverses et Interdisciplinaires – « Défi Modélisation du vivant - 2019», by the GDR ImaBio (AMI fellowship) and by the IFS LyC Collaborative Research Project 2019 (Tohoku University). S Hirose was supported by the STARMAJ Program (France-Japan: Research Internships for Master's Students, Université de Lyon) and K Funamoto by the CNRS (invited researcher position). This project has received funding from the European Research Council (ERC) under the European Union's Horizon 2020 research and innovation program (grant agreement No 639638 to V Calvez) and from Agence Nationale de la Recherche (grant 2019, ADHeC project, No ANR-19-CE45-0002-02 to JP Rieu).

## Additional information

### Funding

| Funder | Grant reference number | Author |
|---|---|---|
| IFS | J19Ly11 | Kenichi Funamoto Jean-Paul Rieu |
| Université de Lyon | STARMAJ | Satomi Hirose |
| European Research Council | 639638 | Vincent Calvez |
| Agence Nationale de la Recherche | ANR-19-CE45-0002-02 | Jean-Paul Rieu |
| GDR Imabio | AMI | Christophe Anjard Jean-Paul Rieu |

| Centre National de la Recherche Scientifique | MITI 2019 | Vincent Calvez Jean-Paul Rieu |

The funders had no role in study design, data collection and interpretation, or the decision to submit the work for publication.

## Author contributions
Olivier Cochet-Escartin, Conceptualization, Data curation, Formal analysis, Supervision, Investigation, Visualization, Methodology, Writing - original draft, Project administration, Writing - review and editing; Mete Demircigil, Conceptualization, Formal analysis, Investigation, Visualization, Methodology, Writing - original draft, Writing - review and editing; Satomi Hirose, Blandine Allais, Formal analysis, Investigation, Methodology; Philippe Gonzalo, Resources, Methodology, Writing - review and editing; Ivan Mikaelian, Resources, Formal analysis, Methodology; Kenichi Funamoto, Conceptualization, Resources, Formal analysis, Supervision, Investigation, Methodology, Project administration, Writing - review and editing; Christophe Anjard, Conceptualization, Resources, Formal analysis, Supervision, Investigation, Methodology, Writing - original draft, Project administration, Writing - review and editing; Vincent Calvez, Conceptualization, Supervision, Funding acquisition, Validation, Investigation, Methodology, Writing - original draft, Project administration, Writing - review and editing; Jean-Paul Rieu, Conceptualization, Data curation, Formal analysis, Supervision, Funding acquisition, Investigation, Visualization, Methodology, Writing - original draft, Project administration, Writing - review and editing

## Author ORCIDs
Olivier Cochet-Escartin (iD) https://orcid.org/0000-0001-7924-702X
Satomi Hirose (iD) http://orcid.org/0000-0002-8594-8006
Philippe Gonzalo (iD) http://orcid.org/0000-0002-9763-0150
Kenichi Funamoto (iD) https://orcid.org/0000-0002-0703-0910

## Decision letter and Author response
Decision letter https://doi.org/10.7554/eLife.64731.sa1
Author response https://doi.org/10.7554/eLife.64731.sa2

# Additional files

## Supplementary files
• Source data 1. Detailed measurements for quantities presented in the text as means ± standard deviations.

• Transparent reporting form

## Data availability
All data generated or analysed during this study are included in the manuscript and supporting files.

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
