## [Decision Letter]

**Acceptance summary:**

The manuscript describes a very interesting novel observation of the collective response of cells of cellular slime mold *Dictyostelium* discoideum: it forms dense rings in response to oxygen gradients. Using carefully executed experimental work combined with thorough numerical and mathematical modeling the authors plausibly show that the phenomenon can be explained through a combination of concentration-dependent taxis to oxygen gradients, and increased motility at low oxygen concentrations. The work is of interest to developmental biologists and mathematical biologists, as well as to cancer researchers due the similarity to cancer cell migration under hypoxia.

**Decision letter after peer review:**

Thank you for submitting your article "Hypoxia triggers collective aerotactic migration in *Dictyostelium* discoideum" for consideration by *eLife*. Your article has been reviewed by 2 peer reviewers, and the evaluation has been overseen by a Reviewing Editor and Naama Barkai as the Senior Editor. The following individuals involved in review of your submission have agreed to reveal their identity: Roeland MH Merks (Reviewer #1); Robert Insall (Reviewer #2).

Summary:

The manuscript describes a highly interesting novel observation of the collective response of cells of cellular slime mold *Dictyostelium* discoideum: it forms dense rings in response to oxygen gradients. Using experimental work combined with numerical and mathematical modeling the authors plausibly show that the phenomenon can be explained through a combination of concentration-dependent taxis to oxygen gradients, and increased motility at low oxygen concentrations. The findings are relevant to a surprisingly wide range of biomedicine – especially tumour growth, but also carnivore foraging (particularly amoebas). However, the biological relevance of the phenomenon is not clear and the model predictions are not evaluated experimentally. The major conclusions are largely justified, but there is little insight in the correctness or explanatory power of the model predictions. With one or a few simple validation experiments, the model predictions in Figure 5 could probably be tested. This, or even an in-depth discussion of the implications of the model predicts would strengthen the support for the major conclusions. In particular this would complete the paper, which currently seems to divert a little into every more detailed and thorough modeling.

Essential revisions:

Strong points of the work are the interesting novel phenomenon of aerotaxis and aerokinesis in Dd, the thorough quantitative data, including quantitative characterization of the oxygen gradients and cell velocities, as well as the detailed mathematical modeling. However, there are some weaknesses that need to be addressed:

1) The biological relevance of the aerotactic behavior is not discussed. Is ring formation observed under more 'natural' laboratory conditions, or is ring formation a consequence of the aerotactic and aerokinetic behavior that is relevant under other biological conditions? At what life stage of Dd is the behavior important: the single cell or collective stage? Is ring formation true 'collective behavior' like fruiting body formation or e.g. thermo- or phototactic behavior of slugs, or should one see it as a side effect of many individual cells performing the same single-cell behavior together under unusual laboratory conditions (as the only interaction between cells for this phenomenon is through the oxygen levels in the microenvironment). Or can it be considered a model system, e.g., for cancer cell migration under hypoxic conditions.

2) The authors first present a cell-based model, and then they study a "mean field" model that is based on the same underlying biological assumptions as the cell-based model. Interestingly, however, the cell-based model explains the data well, whereas for the mean-field model additional assumptions (a second threshold below which no aerotaxis occurs or alternative response functions) are required. What explains this discrepancy between the two models? We would expect that the mean-field model has the same overall behavior as the cell-based model upon which it is based, unless the assumptions differ or the discrete interactions between cells introduce new dynamics. For example, one could imagine that cell adhesion represented in the CPM strengthens the formation of the ring (i.e. reduces cell diffusion at higher cell densities). Could this or other "hidden" features of the CPM be introduced into the PDE and suffice to explain the discrepancy? What is the effect of the CPM parameters on ring formation?

3) The cell-based models and mathematical model produce interesting predictions as outlined in Figure 5 and in the expressions of wave speed for example. Yet the modeling is an end point. The models are compared with the go-or-grow models and with the Fisher-KPP reaction-diffusion-model, but the predictions are not used to further analyse the system, which makes the story not 'round' – the modeling seems to get ever more detailed until it reaches a sort of 'dead end'. Yet many of the predictions can be tested. E.g. divisions (Figure 5B) can be inhibited. The oxygen gradients can be manipulated. Motility (D) can be changed etc.

4) The authors are making conclusions about "aerokinesis" because cells experiencing a gradient but behind the crest are highly motile. Another explanation is that the motile cells' behaviour is due to different (greater?) resolvable directional information (Tweedy and Endres, Scientific Reports, around 2014). Could the authors perform experiments to test whether cells at the "aerokinetic" O2 levels behave the same way if they are not experiencing a gradient? Can the authors keep cells in a homogeneous O2 at that level and see if they still polarise and move fast? If so it is indeed aerokinesis. Alternatively, the authors could discuss the issue.

5) Around line 497 the authors make some strong conclusions about the generality of the system. But it seems they are all based on a strictly 2D system, and some of the most interesting conditions are nearly 1D (cells moving down a pipe) or 3D (real life colonies of amoebas or bacteria in soil; there's lots of 2D action in soil but O2 is likely freely available in most). Could the authors (a) try a 1D experiment (b) consider a 3D thought experiment or model? In any case they should discuss how the 1D and 3D cases differ and how important they might be. Maybe even semi-3D (thickish layers?).

6) An open request, which would help future work to pick the molecular bones of this work – is for an overt description of the concentrations of O2 that provoke the strongest responses.

*Reviewer #1:*

The manuscript by Cochet-Escartin et al. describes a novel, aerotactic behavior of *Dictyostelium* discoideum (Dd) and analyses in it in great detail using experimental and mathematical tools, leading to compelling quantitative data of the phenomenon. When Dd colonies are covered, they migrate outwards towards high-oxygen conditions in a dense ring. Inside the central zone surrounded by the ring, Dd cells assume a quiescent, elongated phenotype, while within the ring and outside cells are more rounded and migratory. The behavior of the cells was hypothesized to be regulated by oxygen, and therefore the response of the Dd cells was studied inside a microfluidic device allowing tight control of the oxygen gradient. In particular it was found that a low oxygen levels (<2 %) the cells became sensitive to oxygen gradients, where they moved to higher levels of oxygen. After that using an oxygen sensitive substrate, it was found that oxygen gradients are self-generated by Dd cells due to oxygen consumption, leading to the formation of the ring.

Based on these observations, the authors proposed two types of mathematical models. The first model is a cell-based simulation model based on a hybrid Cellular Potts and PDE model. It assumes a field of oxygen that is formed through leakage of oxygen through the polystyrene bottom of and the cells' consumption of oxygen. The cells switch from a quiescent to a motile state if the oxygen level drops below a threshold, at the motile state they become aerotaxic and they proliferate. These assumptions suffice to reproduce the observed ring formation, oxygen gradient and the low density of motile cells in the center of the colony.

The second model is a continuum model based on the same assumptions, leading to a 'go or grow' type reaction-advection-diffusion equation. Aerotaxis again occurs only at low oxygen concentration (i.e. the advection term) and there is an oxygen-dependent proliferation term that lets cells proliferate only at above-threshold oxygen levels. Interestingly the model can be solved in part analytically, leading to an expression for the wave speed that depends on the growth rate as well as the diffusion rate and for sufficiently strong aerotaxis also on the aerotaxis. The numerical simulations do not match the observations well: in particular the concentration of cells in the core of the ring remains much higher than what is observed in the data (e.g. Figure 6C). In order to "fix" this discrepancy between model and data, a further assumption is introduced, namely a second, lower threshold C_0_' below which the cells become insensitive to the gradients. I.e. only in the range [C_0_' – C_0_] do the cells become aerotactic. Also other assumptions lead to a better match with the data.

Strong points of the work are the interesting novel phenomenon of aerotaxis and aerokinesis in Dd, the thorough quantitative data, including quantitative characterization of the oxygen gradients and cell velocities, as well as the detailed mathematical modeling.

However, the biological relevance of the aerotactic behavior is not discussed at all. It is not clear whether ring formation is observed under more 'natural' laboratory conditions, or whether ring formation is a consequence of the aerotactic and aerokinetic behavior that is relevant under other biological conditions. It is also not clear at what life stage of Dd this behavior is important, the single cell or collective stage, and whether ring formation is true 'collective behavior' like fruiting body formation or e.g. thermo- or phototactic behavior of slugs. Or should we see it as a side effect of many individual cells performing the same single-cell behavior together under unusual laboratory conditions? I would tend towards the latter (lab conditions) as the only interaction between cells for this phenomenon is through the oxygen levels in the microenvironment. Or should we really see this as a model system, e.g., for cancer cell migration under hypoxic conditions?

Furthermore, the authors first present a cell-based model, and then they study a "mean field" model that is based on the same underlying biological assumptions as the cell-based model. Interestingly, however, the cell-based model explains the data well, whereas for the mean-field model additional assumptions (a second threshold below which no aerotaxis occurs or alternative response functions) are required. What explains the discrepancy between the two models is not clarified. We would expect that the mean-field model has the same overall behavior as the cell-based model upon which it is based, unless the assumptions differ or the discrete interactions between cells introduce new dynamics.

The cell-based models and mathematical model produce interesting predictions as outlined in Figure 5 and in the expressions of wave speed for example. Yet the modeling is an end point. The models are compared with the go-or-grow models and with the Fisher-KPP reaction-diffusion-model, but the predictions are not used to further analyse the system, which makes the story not 'round' – the modeling seems to get ever more detailed until it reaches a sort of 'dead end'. Yet many of the predictions can be tested. For example, divisions (Figure 5B) can be inhibited, the oxygen gradients can be manipulated and motility (D) can be changed.*Reviewer #2:*

The authors combine wet experiments, cellular automaton modelling and analytical maths in a dazzling way to show that cells collectively run away from hypoxia (that they have collectively generated). This is a really cool paper. It's relevant to a suprisingly wide range of biomedicine – especially tumour growth, but also carnivore foraging (particularly, of course, amoebas). It's in an exciting and visible area of science. In a sense it's obvious, but – after reading this paper. The data and models are mutually supportive and combine to make a thorough and scholarly exploration of the area.The combination of wet experiments, models and math is this paper's strength; bringing different informative figures together to make a clear conclusion. Its weakness to some will be a lack of description of the underlying pathways.

Two areas where there appears to be lack of clarity:

I'm not sure it's wise to conclude aerokinesis (that is, particular O2 concentrations without a gradient promote nondirectional movement) without an explicit test. If I've understood right the cells have not been seen moving faster at low O2 with no gradient; they've just been seen going particularly rapidly at one point in the gradient.

Another explanation (Tweedy and Endres, Scientific Reports, around 2014; the same author as the later self-generated gradient papers) is that the motile cells' behaviour is due to different (greater?) resolvable directional information.

Around l. 497 the authors make some strong conclusions about the generality of the system. But it seems that they are all based on a strictly 2D system, and some of the most interesting conditions are nearly 1D (cells moving down a pipe) or 3D (real life colonies of amoebas or bacteria in soil; there's lots of 2D action in soil but I think O2 is freely available in most).

---

## [Author Response]

Summary:The manuscript describes a highly interesting novel observation of the collective response of cells of cellular slime mold *Dictyostelium* discoideum: it forms dense rings in response to oxygen gradients. Using experimental work combined with numerical and mathematical modeling the authors plausibly show that the phenomenon can be explained through a combination of concentration-dependent taxis to oxygen gradients, and increased motility at low oxygen concentrations. The findings are relevant to a surprisingly wide range of biomedicine – especially tumour growth, but also carnivore foraging (particularly amoebas). However, the biological relevance of the phenomenon is not clear and the model predictions are not evaluated experimentally. The major conclusions are largely justified, but there is little insight in the correctness or explanatory power of the model predictions. With one or a few simple validation experiments, the model predictions in Figure 5 could probably be tested. This, or even an in-depth discussion of the implications of the model predicts would strengthen the support for the major conclusions. In particular this would complete the paper, which currently seems to divert a little into every more detailed and thorough modeling.

We thank the Reviewers and Editors for their comments. In response, many aspects of the papers have been modified to address the concerns raised. We agree that our findings are “relevant to a wide range of biomedicine” questions but that our modelling work was somewhat too focused on our own experiments and, in places, failed to demonstrate their generality. As a result, particular attention was given to this part of the paper which has been almost entirely reworked. We now show that our analytical approach can not only reproduce our own experimental data but give insight into front propagation under a single self-generated gradient regardless of the underlying details of the process. Along the same line, we have also added new experiments and simulations to further test the predictions of our models. The introduction and Discussion section have been significantly reworked to include discussions on the generality and biological relevance of our results. Finally, we now demonstrate the aerokinesis of Dd cells in homogenous environments through new experiments. Below, we offer a point by point response to the essential revisions which recapitulated all concerns raised by the reviewers.

Essential revisions:Strong points of the work are the interesting novel phenomenon of aerotaxis and aerokinesis in Dd, the thorough quantitative data, including quantitative characterization of the oxygen gradients and cell velocities, as well as the detailed mathematical modeling. However, there are some weaknesses that need to be addressed:1) The biological relevance of the aerotactic behavior is not discussed. Is ring formation observed under more 'natural' laboratory conditions, or is ring formation a consequence of the aerotactic and aerokinetic behavior that is relevant under other biological conditions? At what life stage of Dd is the behavior important: the single cell or collective stage? Is ring formation true 'collective behavior' like fruiting body formation or e.g. thermo- or phototactic behavior of slugs, or should one see it as a side effect of many individual cells performing the same single-cell behavior together under unusual laboratory conditions (as the only interaction between cells for this phenomenon is through the oxygen levels in the microenvironment). Or can it be considered a model system, e.g., for cancer cell migration under hypoxic conditions.

We thank the Reviewer for these comments and apologize for not being clearer. We agree that our results are based on an artificial situation under controlled laboratory conditions and do not directly describe a real life situation. However, we demonstrate throughout the paper that this behavior is governed by aerotaxis and self-generated gradients. Our paper demonstrates this ability in a new system, Dd cells. By demonstrating the ability of Dd cells to perform both aerokinesis and aerotaxis, we agree that “ring formation is a consequence of aerotactic and aerokinetic behavior that is relevant under other biological conditions.”

Indeed, aerotaxis is a strategy to avoid unfavorable O_2_ conditions probably conserved among all eukaryotes. It may play a role in placentation, in tumor progression by orienting the migration and invasion of the cells of the primary hypoxic tumor to the blood capillaries, promoting metastatic spread. We believed these examples discussed in the second paragraph of the manuscript introduction demonstrate that aerotaxis is relevant in various physiological situations. We added another interesting notion that gradients of O_2_ and energy metabolism govern spatial patterning in various embryos. This notion dates back to the classic work of Child (Child 1941). Such notions have mostly been abandoned due to inability to visualize such a gradient or clarify whether they are the result or the cause of developmental patterning (Coffman and Denegre 2007). More recently, Chang et al. found an asymmetric distribution of hypoxia-inducible factor regulating dorsoventral axis establishment in the early sea urchin embryo (Chang et al. 2017). Interestingly, they also found evidence for an intrinsic hypoxia gradient in embryos, which may be a forerunner to dorsoventral patterning. They did not measure directly oxygen nor explained the possible origin of this gradient. We believe new tools, experimental and theoretical, such as those developed in this work, are now available to tackle this question. We hence added the following text to the introduction of the paper:

Addition to the text: “Aerotaxis may also play a role in morphogenesis. […] Interestingly, they also found evidence for an intrinsic hypoxia gradient in embryos, which may be a forerunner to dorsoventral patterning.”

Regarding self-generated gradients, besides the case of O_2_, such gradients of different molecules consumed or degraded by cells are increasingly encountered in the literature. We already cited the important contribution of Reviewer #2 in the introduction (paragraph 3) and in the conclusion (paragraph 4). We again insist on the very diverse patterns including rings that have been found for a long time with bacteria when food or the diffusion of some metabolite including O_2_ is limited (J Adler 1966)(Budrene and Berg 1991). Another interesting system is in plants. Negative plant soil feedback also explains ring formation in clonal plants (Carteni et al., 2012). One key aspect of self-generated gradients that is currently emerging is their ability to drive long-lasting migrations in complex environments and to enhance the robustness of collective migrations. These physiologically relevant aspects of self-generated gradients were demonstrated by the team of Reviewer #2 (Tweedy and Insall, 2020, Tweedy et al., 2020) and others (Cremer et al., 2019). We have expanded on this aspect in the introduction of the paper to underline the physiological relevance of collective migrations driven by self-generated gradients.

Addition to the text: “Physiologically speaking, self-generated gradients have been demonstrated to increase the range of expansion of cell colonies (Cremer et al. 2019; Tweedy and Insall 2020) and to serve as directional cues to help various cell types navigate complex environments, including mazes (Tweedy et al. 2020). Oxygen self-generated gradients could therefore play important roles in a variety of contexts, such as development, cancer progression or even environmental navigation in the soil.”

We have also added a few sentences at the beginning of the Discussion section to more directly highlight the possible physiological relevance of our observations.

Addition to the text: “Overall, our results demonstrate the ability of Dd cells to respond to hypoxia through both aerotactic and aerokinetic responses. […] Finally, our work goes beyond these results as it demonstrates that oxygen can play the role of the attractant in self-generated gradients therefore potentially extending the physiological relevance of the use of such cues in collective migration.”

Still, “Natural laboratory conditions” might also mean “close to physiological conditions” and the reviewer might wonder whether Dd cells actually encounter hypoxia in their normal conditions. In the natural habitat of Dd, the soil, depletion of nutrients or of oxygen is generally gradual and growth phase cells have mechanisms to sense when hard times are approaching by secreting factors (Maeda 2005). But, abrupt oxygen changes such as the one induced by our vertical confinement with the coverglass might also happens after a flood (Pedersen, Perata, and Voesenek 2017).

Regarding the question about the life stage of Dicty, we believe that this may indicate that the reviewer has in mind classical Dd development conditions after abruptly washing nutrients and putting cell in a starvation buffer. To approach that question and to reply as well to item #3 below (i.e., about the influence of cell division), we performed our confinement spot assay in a phosphate starvation buffer. A ring indeed develops quickly, in the same timing as in the nutrient buffer HL5 but it quickly dissipates as Dd cells stop dividing in these conditions. Still, this new experiment shows that ring formation happens during both the early development stage (after abrupt starvation) and during the growth phase. Description of this experiment was added at the end of the Potts simulation section of the paper and we added a new figure supplement and a new video to report its results.

Addition to the text: “First, we show in Figure 5B the effect of turning cell division off in the simulated spot. […] This still demonstrates that the phenomenon observed here is relevant for both the single cell and collective stages of Dd life cycle.”

The reviewer also questions whether the observed phenomenon is truly a collective one. We agree that the expansion of the ring is due to “many individual cells performing the same single-cell behavior”. However, we argue that ring formation is a response to a collective stress. The response is clearly collective as well and we can talk about an emergent behavior, because a single confined cell or even a too loose colony will not produce a ring. The ring is the consequence of the O_2_ gradient generated by the high cell density and the limited available O_2_. Said differently, we agree that the “only interaction between cells (…) is through the oxygen levels” but these oxygen levels are themselves the result of the presence of a large population of cells. For these reasons, we have made the choice of describing this phenotype as a collective migration but specifying that it occurs without direct cell-cell communication.

Finally, we also agree with the reviewer that our confined spot assay is a good model system to study the areotaxis of many cell types including cancer cells. Some of the authors developed it on epithelial cells (Deygas et al. 2018) and this manuscript demonstrates the robustness of the readouts (ring speed, shape…) for amoeboid cells. We believe that invasive cells might also be studied under this assay. We have added a short sentence to the end of the first Results subsection to underline this aspect:

Addition to the text: “This shows that the spot assay is an excellent experimental system to study the response of a variety of cell types to vertical confinement and its physiological consequences (Deygas et al. 2018).”

2) The authors first present a cell-based model, and then they study a "mean field" model that is based on the same underlying biological assumptions as the cell-based model. Interestingly, however, the cell-based model explains the data well, whereas for the mean-field model additional assumptions (a second threshold below which no aerotaxis occurs or alternative response functions) are required. What explains this discrepancy between the two models? We would expect that the mean-field model has the same overall behavior as the cell-based model upon which it is based, unless the assumptions differ or the discrete interactions between cells introduce new dynamics. For example, one could imagine that cell adhesion represented in the CPM strengthens the formation of the ring (i.e. reduces cell diffusion at higher cell densities). Could this or other "hidden" features of the CPM be introduced into the PDE and suffice to explain the discrepancy? What is the effect of the CPM parameters on ring formation?

The reviewer is correct in that Potts simulations and the elementary Go or Grow model led to different predictions, especially in terms of the shape of the advancing front. The referee is also correct in pointing out that both approaches are based on the same underlying assumptions. However, one goal of this elementary Go or Grow model was to remain analytically solvable so that we could have a more detailed understanding of key parameters than with Potts simulations. For that reason, the elementary Go or Grow model wasn’t a mean field representation of the Potts model as many details were different. For instance, the aerotactic response was set to be proportional to the local gradient in Potts simulations but based solely on its direction in the elementary Go or Grow model. The form of oxygen consumption by cells was also different in both approaches. Finally, our previous version of the Potts model didn’t have reduced cell division at low O_2_ concentrations unlike the elementary Go or Grow model. Still, we agree with the reviewer that these differences were not clearly put forward and that the discrepancies between the two models were thus problematic. Based on this comment, we have made numerous changes to both the Potts and the Go or Grow section of the paper. In summary:

– Potts simulations now incorporate the fact that cells stop dividing below an O_2_ threshold, a biologically sound hypothesis (Giampietro Schiavo and Bissons 1989; West, van der Wel, and Wang 2007), thus turning it into a Go or Grow model as well. Figures showing the results of these simulations have been updated accordingly.

Addition to the text: “Finally, all cells can divide as long as they sit in a high enough O_2_ concentration since it was demonstrated that cell division slows down in hypoxic conditions (G. Schiavo and Bisson 1989; West, van der Wel, and Wang 2007)”.

– The mathematical section of the paper now starts with a mean field representation of these Potts simulations. We show that this analytical approach can therefore recapitulate the numerical simulations when all ingredients are similar and hence recapitulate with a very satisfactory agreement the experimental data.

– Since this mean field approach is not solvable, we then turn to our original elementary Go or Grow model which gets further away from our experiments but allows us to make quantitative, general predictions in particular about the wave speed, its independence towards oxygen dynamics and the relative role of division and aerotaxis in setting the wave speed.

– We now end by showing variations on this Go or Grow model not in order to reproduce our experiments but to show the applicability of our method and to underline the fact that the analytical predictions are robust with respect to small modifications of the model, hence showing the general relevance of our results beyond the aerotaxis of Dd cells.

As a result, the mathematical section of the model has been largely rewritten and re-organized. Figures 6 and 7 have been modified to include this new structure. One issue we pointed out with the Potts model (formation of a dense core of cells) was removed as it no longer is true and this actually improves the quality of these simulations. Finally, Methods sections have been modified to reflect these changes.

We thank the reviewer for this comment as we believe it has made our approach more sound and easier to follow for the reader while insisting more on general results rather than simply reproducing the experiments. In that sense, these changes also partly address Comment #3 below.

3) The cell-based models and mathematical model produce interesting predictions as outlined in Figure 5 and in the expressions of wave speed for example. Yet the modeling is an end point. The models are compared with the go-or-grow models and with the Fisher-KPP reaction-diffusion-model, but the predictions are not used to further analyse the system, which makes the story not 'round' – the modeling seems to get ever more detailed until it reaches a sort of 'dead end'. Yet many of the predictions can be tested. E.g. divisions (Figure 5B) can be inhibited. The oxygen gradients can be manipulated. Motility (D) can be changed etc.

We agree with the reviewer that both our models led to interesting predictions that were not tested experimentally to further justify their generality.

We have now incorporated two such comparisons between the Potts simulations and experiments.

First, we performed spot assay experiments in phosphate buffer (see response to comment #1). This experiment not only demonstrate that ring formation also occurs under starvation conditions, it allows us as well to test the behavior of this ring in the absence of cell division (see figure 5—figure supplement 2). The agreement with the simulations (Figure 5B) is very satisfactory as the ring first propagated but quickly stopped after 3h and finally dispersed as cell density was no longer sufficient to maintain hypoxic conditions.

In addition, we also asked what would be the behavior of rings under more complex oxygen environments. As a first step, we asked what would be the behavior emerging from two colonies put in close proximity. Would the rings repel each other, attract each other, avoid each other? There too, we found good qualitative agreement between the simulations and experiments: these rings will fuse, form an oblong shape that will relax towards a circular one thus forming one large ring as a result. We have added a new figure supplement and a new video to document these two results. We have also added the paragraph below to the end of the Potts section of the paper to introduce these tests.

Addition to the text:

“Because of their versatility, they can also be used to make some predictions on the observed phenomenon. […] However, they fall short of giving an in-depth quantitative description because they rely on many parameters and are not amenable to theoretical analysis per se.”

We also agree that our analytical efforts reached a ‘dead end’ and we believe this was due to our attempt at tuning this model to simply reproduce our experiments. As described in response to comment #2, we have now completely restructured this part of the paper and focused on the applicability of our results to different systems and experiments rather than focusing solely on our own.

– The mathematical section of the paper now starts with a mean field representation of these Potts simulations. We show that this analytical approach can therefore recapitulate the numerical simulations when all ingredients are similar.

– Since this mean field approach is not solvable, we then turn to our original elementary Go or Grow model which gets further away from our experiments but allows us to make quantitative, general predictions in particular about the wave speed, its independence towards oxygen dynamics and the key role of division and aerotaxis.

– We now end by showing variations on this Go or Grow model to show the applicability of our method and to underline the fact that the analytical predictions are robust with respect to small modifications of the model, hence showing the general relevance of our results beyond the aerotaxis of Dd cells.

We agree that other comparisons between predictions of the models and experiments would be beneficial but many are difficult to achieve and we feel go beyond the scope of this paper. In the near future, we intend to pursue the fine manipulation of oxygen gradients and oxygen levels in parallel to refine the aerotactic response of Dd cells.

4) The authors are making conclusions about "aerokinesis" because cells experiencing a gradient but behind the crest are highly motile. Another explanation is that the motile cells' behaviour is due to different (greater?) resolvable directional information (Tweedy and Endres, Scientific Reports, around 2014). Could the authors perform experiments to test whether cells at the "aerokinetic" O2 levels behave the same way if they are not experiencing a gradient? Can the authors keep cells in a homogeneous O2 at that level and see if they still polarise and move fast? If so it is indeed aerokinesis. Alternatively, the authors could discuss the issue.

We thank the reviewer for this remark and we agree that our claim about aerokinesis was based solely on data where cells were also experiencing an oxygen gradient. We thus followed the reviewer’s advice and checked for aerokinesis in homogenous environments.

We built a glass environmental chamber compatible with videomicroscopy studies at very low oxygen concentrations in particular with the random motility assay we have performed for many years in the laboratory (d’Alessandro et al. 2018; Golé et al. 2011). Our current chamber and gas blender enables the control of O_2_ down to 0.4%. Cell diffusion constant was very significantly higher at 0.4% (D=40.2±9.6 µm^2^/min) than at the 20.95 % control condition (D=19.2±8.8 µm^2^/min, Figure 2figure supplement 8 A-D). We could not notice any difference between the diffusion constant of cells that stayed 3h or 20h at 0.4% (Figure 2—figure supplement 8) while control cell motility tends to decrease with time as quorum sensing factors accumulates in the medium (d’Alessandro et al. 2018; Golé et al. 2011). Noticeably, cell trajectories were more homogeneous at 0.4% than at 20.95%. A reason could be that cells almost stop dividing at 0.4% while the motility of control cells might be affected by cell-cycle position. About cell shape and cell polarization, we did not image classical polarization markers but this is something we plan to do for future studies. However, the shapes of the cells in the hypoxic condition are more elongated than in normoxic conditions (not shown but similar to the shape of cells respectively at the left, i.e., hypoxic region, and right side, i.e., normoxic region, of the ring in Figure 1C). All together, we showed for the first time, to our best knowledge, an aerokinetic effect in *Dd* at very low O_2_ concentrations. This work offers very interesting perspective for future work to measure motility and phenotype changes as a function of oxygen level in the 0.1%2% range where we observe an aerotactic response. We have added a figure supplement to Figure2 to recapitulate these results along with a description of these experiments into the microfluidics section of the text.

Addition to the text: “Since cells in the microfluidic devices were also experiencing oxygen gradients, we further tested if the observed was true aerokinesis. […] The very significant difference (p<0.0001) demonstrates that Dd cells show an aerokinetic positive response to low oxygen, even in the absence of gradients.”

5) Around line 497 the authors make some strong conclusions about the generality of the system. But it seems they are all based on a strictly 2D system, and some of the most interesting conditions are nearly 1D (cells moving down a pipe) or 3D (real life colonies of amoebas or bacteria in soil; there's lots of 2D action in soil but O2 is likely freely available in most). Could the authors (a) try a 1D experiment (b) consider a 3D thought experiment or model? In any case they should discuss how the 1D and 3D cases differ and how important they might be. Maybe even semi-3D (thickish layers?).

We agree that this comment deserves a short discussion in the manuscript and we thank the reviewer to point this lack. Our spot assay is indeed almost purely 2D as the vertical confinement thickness e~50µm is negligible as compared to radial directions. We also agree that in soil the geometry is probably hybrid with cells restricted to move on complex surfaces (grains or soil fibers) but with O_2_ diffusing from the whole surrounding space in 3D. As such, our experiments are clearly simplifications of the environment actually encountered by Dd cells in the wild.

However, our experimental measurements as well as our numerical simulations show that an important property of the migrating front is the fact that it follows a constant oxygen concentration within the dynamic gradients. This result doesn’t depend on the dimensionality of the problem and simply requires that hypoxic conditions are reached and that self-generated gradients are present, i.e. as long as cell density is high enough and oxygen sources are further away from the cells. For instance, a similar behavior in different geometries was already observed by Reviewer #2 using the under agarose assays (Tweedy and Insall 2020). The ~1-mm thick agarose layer soaked with folate restricts convection without greatly affecting diffusion. The reviewer shows that self-generated gradients of degraded folate induce a group migration of cells in bands (in 1D) or rings (in 2D spots) up to 4 mm. This work, just as ours, points to the importance of a sufficient cell density in the initial colony to degrade enough of the attractant and of cell division to maintain this density throughout the migration over long distances. We therefore believe that our results can be generalized to any dimension and that knowledge of the oxygen profiles arising from cell consumption and diffusion from a source are sufficient to predict the existence and behavior of a migrating front.

In 3d: under spherical symmetry, we expect the formation and migration of a dense spherical ring of cells. The cell density required to achieve hypoxic conditions could however differ from 2d situations because diffusion will be more efficient at replenishing the oxygen consumed by the cells for geometrical reasons.

In more complex geometries, such as the soil, we believe it is difficult to speculate as to the distribution and diffusion of oxygen in such a complex environment but we believe that a dense enough colony to create hypoxic conditions will trigger the migration of a dense front whose geometry will depend on the shape of the complex oxygen profiles.

Still, to fully demonstrate the generality of our results, we have now added Potts model numerical simulations in 3d, one showing the formation of a dense sphere of cells when oxygen sources are present in all directions. In a second one, we model a slightly more complex geometry in which oxygen sources are present on all sides and on the top of the original colony but not on the bottom. We then observe the migration of a half-spheric front moving upwards towards the source but no front was observed moving downwards. Of note, for computational reasons, these simulations were made using a minimal Potts model that is no longer quantitative but possesses all the key ingredients described in the text. We have added a paragraph in the Discussion section addressing this role of dimensionality and introducing these new results which we present as an additional video.

Concerning the analytical results, under the assumption of spherical symmetry, the equation reduces to the 1D case when curvature effects can be neglected. This is actually the natural mathematical framework to construct traveling wave solutions which are stationary in the moving frame. Indeed, curvature transiently slow down the speed of propagation as compared to the 1D case. Therefore, the 1D analysis can be viewed as the long time asymptotic of planar front propagation beyond the curvature effects.

Addition to the text:

“In addition, although our experimental results were obtained on simple, 2d experiments, our findings can generalize to more complex cases. […] Our 2d results can therefore be extended to any other situations and they show that the key to proper steering are high enough cell densities and the creation of robust self-generated gradients.”

6) An open request, which would help future work to pick the molecular bones of this work – is for an overt description of the concentrations of O2 that provoke the strongest responses.

We thank the reviewers and editor for this suggestion to which we clearly adhere. We believe it also echoes one of the reviewers’ comment: “The combination of wet experiments, models and math is this paper's strength; bringing different informative figures together to make a clear conclusion. Its weakness to some will be a lack of description of the underlying pathways.”

In our opinion, our work calls for two main prolongations in the future:

– An in-depth exploration of the response of individual Dd cells to more diverse oxygen environments. For instance, as described in the response to comment #3, we will now focus on developing tools to independently control oxygen levels and gradients to better study the crosstalk between these two aspects, crosstalk which we find to be central to the collective behavior we observed. For instance, it would be very informative to observe Dd cells in the 02% range of oxygen where they are aerotactic and tune the steepness of the gradient within this range to pinpoint the exact aerotactic modulation by local concentrations.

– A molecular characterization of the reponse of Dd cells to hypoxia and oxygen gradients. The spot assay which is easy to use experimentally could be used in this context as a screening tool for mutants.

Although we fully agree that these are natural steps stemming from this paper, we also feel like they both require long developments and/or experiments and that they would produce data which could be drowned in this paper that already touches on many different fields and aspects of the problem. Both would probably be better suited for systematic analysis.

References:

Adler, J. 1966. “Chemotaxis in Bacteria.” Science 153: 708.

Adler, Julius. 1966. “Chemotaxis in Bacteria.” Science 153(3737): 708–16.

https://pubmed.ncbi.nlm.nih.gov/4957395/ (December 9, 2020).

Budrene, Elena O, and Howard C Berg. 1991. “Complex Patterns Formed by Motile Cells of *Escherichia coli*.” 349(February): 630–33.

Chang, Wei-lun et al. 2017. “Asymmetric Distribution of Hypoxia-Inducible Factor α Regulates Dorsoventral Axis Establishment in the Early Sea Urchin Embryo.” Development 144: 2940–50.

Child, C. M. 1941. “Formation and Reduction of Indophenol Blue in Development of an Echinoderm.” Proceedings of the National Academy of Sciences 27(11): 523–28. https://www.ncbi.nlm.nih.gov/pmc/articles/PMC1078374/ (April 8, 2021).

Coffman, J.A., and J.M. Denegre. 2007. “Mitochondria, Redox Signaling and Axis Specification in Metazoan Embryos.” Dev Biol. 308: 266–80.

Cremer, Jonas et al. 2019. “Chemotaxis as a Navigation Strategy to Boost Range Expansion.” Nature 575(7784): 658–63. https://doi.org/10.1038/s41586-019-1733-y (April 8, 2021). d’Alessandro, Joseph et al. 2018. “Collective Regulation of Cell Motility Using an Accurate DensitySensing System.” Journal of The Royal Society Interface 15(140): 20180006. https://royalsocietypublishing.org/doi/10.1098/rsif.2018.0006 (June 5, 2020).

Deygas, Mathieu et al. 2018. “Redox Regulation of EGFR Steers Migration of Hypoxic Mammary Cells towards Oxygen.” Nature communications 9(1): 4545. http://www.nature.com/articles/s41467018-06988-3 (December 21, 2018).

Golé, Laurent, Charlotte Rivière, Yoshinori Hayakawa, and Jean Paul Rieu. 2011. “A Quorum-Sensing Factor in Vegetative *Dictyostelium* Discoideum Cells Revealed by Quantitative Migration Analysis.” PLoS ONE 6(11).

Gregor, Thomas, Koichi Fujimoto, Noritaka Masaki, and Satoshi Sawai. 2010. “The Onset of Collective Behavior in Social Amoebae.” Science 328(5981): 1021–25.

Kelly, Beth et al. 2021. “Sulfur Sequestration Promotes Multicellularity during Nutrient Limitation.” Nature.

Maeda, Yasuo. 2005. “Regulation of Growth and Differentiation in *Dictyostelium*.” International Review of Cytology 244: 287.

Pedersen, O., P. Perata, and L. Voesenek. 2017. “Flooding and Low Oxygen Responses in Plants.” Functional Plant Biology 44: iii–vi.

Saragosti, J et al. 2011. “Directional Persistence of Chemotactic Bacteria in a Traveling Concentration Wave.” Proceedings of the National Academy of Sciences 108(39): 16235–40. http://www.pnas.org/cgi/doi/10.1073/pnas.1101996108.

Schiavo, G., and R. Bisson. 1989. “Oxygen Influences the Subunit Structure of Cytochrome c Oxidase in the Slime Mold *Dictyostelium* Discoideum.” The Journal of biological chemistry 264(13): 7129–34. http://www.jbc.org/article/S0021925818832112/fulltext (April 8, 2021).

Schiavo, Giampietro, and Roberto Bissons. 1989. “Oxygen Influences the Subunit Structure of Cytochrome c Oxidase in the Slime Mold *Dictyostelium* Discoideum ”.” c.

Tweedy, Luke et al. 2020. “Seeing around Corners: Cells Solve Mazes and Respond at a Distance Using Attractant Breakdown.” Science 369(6507). https://science.sciencemag.org/content/369/6507/eaay9792 (April 8, 2021).

Tweedy, Luke, and Robert H. Insall. 2020. “Self-Generated Gradients Yield Exceptionally Robust Steering Cues.” Frontiers in Cell and Developmental Biology 8. /pmc/articles/PMC7066204/?report=abstract (July 9, 2020).

West, Christopher M., Hanke van der Wel, and Zhuo A. Wang. 2007. “Prolyl 4-Hydroxylase-1 Mediates O2 Signaling during Development of *Dictyostelium*.” Development 134(18): 3349–58. http://www.dictybase.org (April 8, 2021).